# In-line swimming dynamics revealed by fish interacting with a robotic mechanism

Robin Thandiackal*, George Lauder*

Harvard University, Cambridge, United States

**Abstract** Schooling in fish is linked to a number of factors such as increased foraging success, predator avoidance, and social interactions. In addition, a prevailing hypothesis is that swimming in groups provides energetic benefits through hydrodynamic interactions. Thrust wakes are frequently occurring flow structures in fish schools as they are shed behind swimming fish. Despite increased flow speeds in these wakes, recent modeling work has suggested that swimming directly in-line behind an individual may lead to increased efficiency. However, only limited data are available on live fish interacting with thrust wakes. Here we designed a controlled experiment in which brook trout, *Salvelinus fontinalis*, interact with thrust wakes generated by a robotic mechanism that produces a fish-like wake. We show that trout swim in thrust wakes, reduce their tail-beat frequencies, and synchronize with the robotic flapping mechanism. Our flow and pressure field analysis revealed that the trout are interacting with oncoming vortices and that they exhibit reduced pressure drag at the head compared to swimming in isolation. Together, these experiments suggest that trout swim energetically more efficiently in thrust wakes and support the hypothesis that swimming in the wake of one another is an advantageous strategy to save energy in a school.

## Editor's evaluation

Why do fish school together? Energetic benefits have long been considered a key factor in motivating fish to swim together and tune their tailbeat to exploit the whirling wake generated by conspecifics. This study clearly demonstrates that fish benefit from swimming in a two-dimensional vortical wake by locating their body in the vortical low-pressure zones that passively impart a net thrust force on their oscillating bodies. The behavioural and biofluid mechanical findings will interest comparative biomechanists, movement ecologists, evolutionary biologists, fluid mechanists, and bioinspired roboticists.

## Introduction

Individuals in fish schools have long been hypothesized to benefit from hydrodynamic advantages associated with swimming near other conspecifics (*Becker et al., 2015*; *Li et al., 2021*; *Park and Sung, 2018*; *Weihs, 1973*). Recent work supports this hypothesis on the basis of experiments where schooling fish exhibit reduced tail-beat frequencies relative to solitary individuals which suggests decreased energy consumption by the group as a whole (*Ashraf et al., 2017*; *Marras et al., 2015*). A number of specific mechanisms have been proposed and investigated to show how corresponding hydrodynamic effects could contribute to reduced energy demands in schools (*Figure 1*). The phalanx or soldier formation describes fish swimming side-by-side, parallel to each other (*Figure 1A*), and fish in this position are expected to benefit from the channeling/wall effect and simulation studies *Daghooghi and Borazjani, 2015*; *Hemelrijk et al., 2015* have shown increased efficiency for this formation. And *Ashraf et al., 2017* linked the phalanx formation to reduced energy consumption in red nose tetras swimming in a school. Another beneficial interaction can occur when two fish swim in close proximity to one another (*Figure 1B*). Here, the leading swimmer is thought to experience

*For correspondence:
rthandiackal@fas.harvard.edu
(RT);
glauder@oeb.harvard.edu (GL)

Competing interest: The authors declare that no competing interests exist.

**eLife digest** Some species of fish swim together in groups known as schools. This behaviour makes it easier to find food, avoid predators, and maintain social interactions. In addition, biologists also think that being in a group reduces the energy needed to swim compared to being alone.

Similar to the tracks that follow ships moving through water, fish also leave a wake behind them as they swim. By flapping their tail side-to-side, they create characteristic patterns in the water, including swirling currents. Fish in a school encounter many of these wakes from their neighbours, and may use this to position themselves relative to each other. Previous studies have suggested that swimming directly behind each other increases swimming efficiency; however, this was based on computer models and experiments on flapping systems rather than real-life settings.

To better understand how swimming in a line works in practice, Thandiackal and Lauder tested this idea in live fish. A robotic flapping foil designed to imitate the tail fin of a leading fish was placed in front of a single trout swimming in a tank with flowing water. The fish positioned itself directly behind the foil and timed its own flapping to match it. The trout also interacted with the swirling currents, which Thandiackal and Lauder calculated helped reduce the resistance from the water flow.

These results suggest that swimming directly behind each other can improve swimming efficiency, complementing previous studies showing the benefits of other formations, such as swimming side-by-side. This suggests that fish in schools may have many opportunities to save energy. In the future, this improved understanding could help to design underwater vehicles that work more efficiently in groups.

increased thrust because of the additional effective added mass at the tail trailing edge due to blockage of water by the trailing swimmer behind. Simulations on pitching foils *Bao and Tao, 2014*; *Saadat et al., 2021* have confirmed this effect and show increased overall hydrodynamic efficiency for the two-body system of leading and trailing swimmers. Measurements of reduced tail-beat frequencies of fish swimming at the front of schools of gray mullet compared to swimming in isolation further support these findings (*Marras et al., 2015*).

A third commonly proposed schooling arrangement is the diamond or staggered pattern (*Figure 1C*) first suggested by *Weihs, 1973*. The value of swimming in this formation is due to the nature of thrust wake vortical structures generated behind swimming fish. Fish thrust wakes are characterized by both a vortex street of alternating orientation, and an increased average flow speed compared to the free stream (*Blickhan et al., 1992*; *Müller et al., 1997*; *Nauen and Lauder, 2002*; *Tytell, 2010*). Weihs hypothesized that fish directly behind another *would experience a higher relative velocity and would have to exert extra energy* and suggested that the most efficient swimming position lies midway between two preceding fish (*Figure 1C*) resulting in a diamond formation. A fish swimming in this diamond formation encounters flow conditions resembling a von Kármán drag wake, similar to the one shed by a cylinder under sufficiently high flow speeds. *Liao et al., 2003* explored this scenario in trout and found reduced muscle activity for fish swimming in a drag wake, and direct measurements of energy consumption confirm that fish experience reduced energetic costs when in a drag wake (*Taguchi and Liao, 2011*).

In contrast to Weihs' argument that in-line fish positions are disadvantageous (*Figure 1D*), some recent work suggests that swimming in tandem provides hydrodynamic advantages. Simulations (*Hemelrijk et al., 2015*; *Maertens et al., 2017*), flapping foil experiments (*Boschitsch et al., 2014*; *Kurt and Moored, 2018*), and robot experiments (*Saadat et al., 2021*) indicate increased thrust production and efficiency when a fish or flapping foil swims in a thrust wake. The fluid dynamic benefits to the follower occur because the swimmer in the thrust wake experiences the oncoming flow at its leading-edge with an oscillating angle of attack and is subject to lift forces that have components in forward direction. Maertens et. al (*Maertens et al., 2017*) argue that a downstream swimmer *can reduce its drag by consistently turning its head in a manner that employs the oncoming vortex flow to increase the transverse velocity across the head.* As a result, the pressure drag at the head can be decreased substantially and result in increased efficiency.

Although recent modeling work suggests advantages for in-line swimming, experimental data on live fish exploiting these conditions is lacking. Do live fish actually take positions in a thrust wake when

**Figure 1.** Schooling positions with hydrodynamic benefits. (**A**) Swimming side-by-side can increase thrust and efficiency by making use of the channeling effect (*Ashraf et al., 2017*; *Daghooghi and Borazjani, 2015*). (**B**) Leading swimmers benefit from higher thrust production due to increased effective added mass at their trailing edge stemming from the blockage of the water in close proximity to trailing swimmers (*Bao and Tao, 2014*; *Saadat et al., 2021*). (**C**) Trailing fish face reduced oncoming flows between two leading fish when swimming in a diamond formation (4). (**D**) Leading-edge suction provides propulsive thrust for a fish in a trailing position (*Kurt and Moored, 2018*; *Maertens et al., 2017*; *Saadat et al., 2021*).

free to swim at any location in flow? When fish swim directly behind another, do they alter their swimming kinematics and is there evidence for a reduction of swimming cost even when in a thrust wake with accelerated mean flow? Here we explore how fish interact with thrust wakes in a controlled experimental setting. We chose trout (brook trout, *Salvelinus fontinalis*) for our investigation as this species swims against oncoming currents in their natural habitat and is known to sense and take advantage of flow structures that can reduce energy use (*McLaughlin and Noakes, 1998*; *Shuler et al., 1994*). Fish moving in fluids use (1) vision, (2) the lateral line, and (3) the vestibular system to control their body motion. All of them have been the subject of numerous studies over the years (*Ali, 2013*; *Coombs and Montgomery, 2014*; *Platt, 1973*). The individuals in our experiments had all of these sensor modalities available. In our approach we emulate the thrust wakes from leading swimmers using an actuated flapping foil that serves as the artificial counterpart of a fish tail-fin. Similar approaches have been proposed in previous work to study attraction of fish to robots (*Marras and Porfiri, 2012*; *Polverino et al., 2013*) and how fish respond to thrust wakes (*Harvey et al., 2022*; *Zhang et al., 2019*). Using a flapping foil allowed us to generate accelerated flows with similar hydrodynamic characteristics, in terms of the Strouhal number and the relative axial and lateral spacing of shed vortices, to those of live fish (*Anderson*

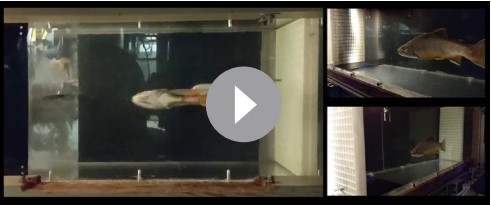

**Video 1.** Trout swimming in the thrust wake of a flapping foil (bottom and side views).
https://elifesciences.org/articles/81392/figures#video1

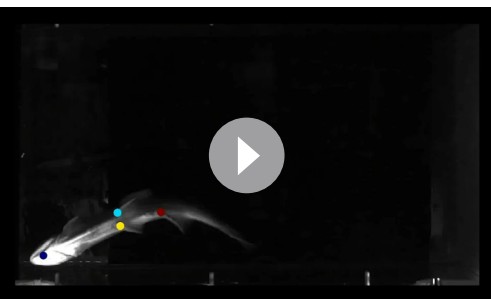

**Video 2.** Time lapse of a trout exploring the flow tank with a thrust wake present (bottom view). Over time trout position themselves in the thrust wake and synchronize with the flapping foil.

https://elifesciences.org/articles/81392/figures#video2

*et al., 1998*; *Buchholz and Smits, 2005*). By carefully choosing the robotic flapping motion, we generated fish-like thrust wakes and introduced trout to these conditions. We found that trout swim in-line with the flapping foil (*Videos 1 and 2*) and reduce their tail-beat frequencies compared to swimming at the same effective flow speeds under free-stream conditions. Further analyses employing particle image velocimetry revealed that individuals interact directly with oncoming thrust wake vortices. Finally, our pressure field computations showed reduced average pressures at the leading-edge suggesting reduced pressure drag and reduced swimming costs. These findings support the hypothesis that fish can reduce swimming costs under in-line swimming conditions and help explain why in-line swimming is common in schools of fish.

## Results

### Reduced frequency and synchronization with a flapping foil

Artificial thrust wakes were generated in a recirculating flow tank using an actuated flapping foil with 2 degrees of freedom which enabled side-to-side movement as well as rotation ('Materials and

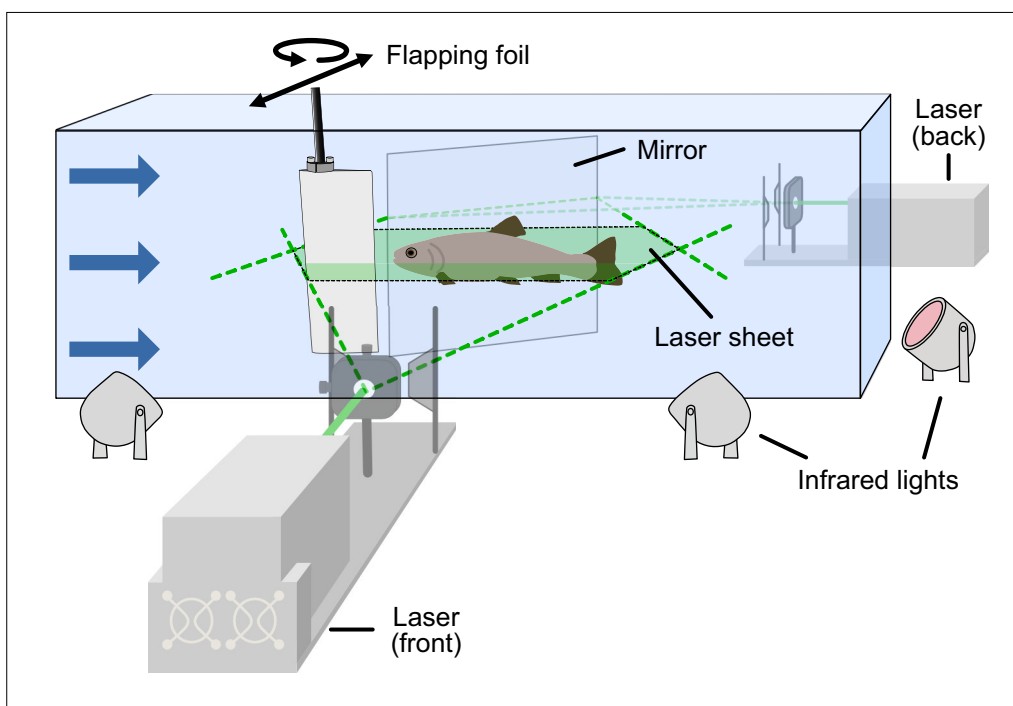

**Figure 2.** Experimental setup. Flapping foil with 2 degrees of freedom (yaw and sway) generating a fish-like thrust wake in the flow tank. Trout swam in the dark while we captured the kinematics by means of high-speed cameras from a bottom and side view and using infrared lights for illumination. Low light in the tank upstream of the flapping foil allowed fish to orient. In separate experiments, we captured the flow dynamics using particle image velocimetry. We were able to record the entire flow field around the fish by using two lasers (in front and behind) simultaneously.

The online version of this article includes the following figure supplement(s) for figure 2:

**Figure supplement 1.** Comparison of flapping foil wake and fish wake.

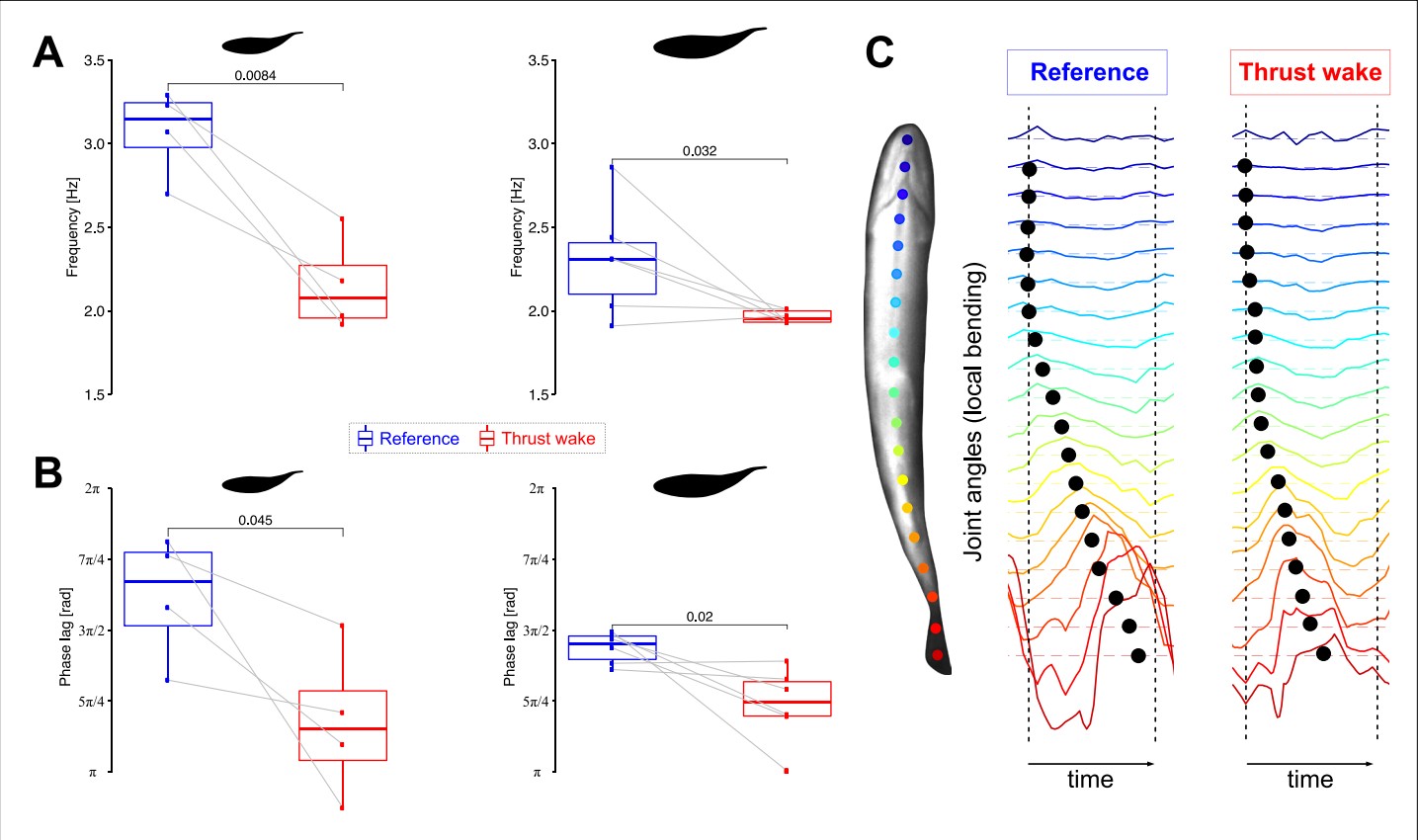

**Figure 3.** Body kinematics in thrust wakes. (**A**) Reduced tail-beat frequencies and (**B**) reduced overall phase lags for small (n = 4) and large (n = 6) trout swimming in the thrust wake compared to steady swimming at the same flow tank speed. (**C**) Illustration of the bending pattern by means of joint angles (rainbow colored lines) along the body. Black markers indicate the bending phase.

The online version of this article includes the following figure supplement(s) for figure 3:

**Figure supplement 1.** Head and tail amplitudes.

methods: Flapping foil,' *Figure 2*). The motion of the foil together with the flow speed ($St = 0.267$) were chosen such that the Strouhal number falls in the typical range of 0.2–0.4 for swimming fish (*Saadat et al., 2017*). The thrust wake generated by the flapping foil is characterized by a reverse Kármán vortex street and increased flow speeds in the wake (*Figure 2—figure supplement 1*) comparable to the wakes generated by swimming trout (*Nauen and Lauder, 2002*). Matching the Strouhal number of swimming fish and our flapping foil ensures similar hydrodynamics in terms of the relative axial and lateral spacing between vortices. It is worth noting that the relatively large span of the flapping foil induces thrust wakes along a larger depth and thus increases the chance that fish encounter the wake in the flow tank, however at the expense of producing two-dimensional (planar) thrust wakes.

We used a paired experimental design and had the same individuals swim under two conditions: in a flow tank with (1) an actuated flapping foil generating a thrust wake, and (2) under control free stream conditions with the foil held in a stationary position in the water. In both conditions, the flow was fixed at the same speed, and thus permitted a controlled comparison of the corresponding swimming patterns. In addition, we carried out the same experiments 2.5 months apart, which allowed us to investigate how differences in body size affect the behavior under the different conditions as fish were larger in total length after this growth period. We captured the swimming kinematics using high-speed video recordings from the ventral perspective and extracted body midlines ('Materials and methods: Experimental setup and Kinematic analysis').

We found that trout from both size groups significantly reduced their tail-beat frequencies when they were exposed to thrust wakes (*Figure 3A*, *Video 2*). Smaller fish showed a decrease of 28.3%, and larger fish showed a decrease of 14.7% in the mean frequency. This suggests that fish maintained

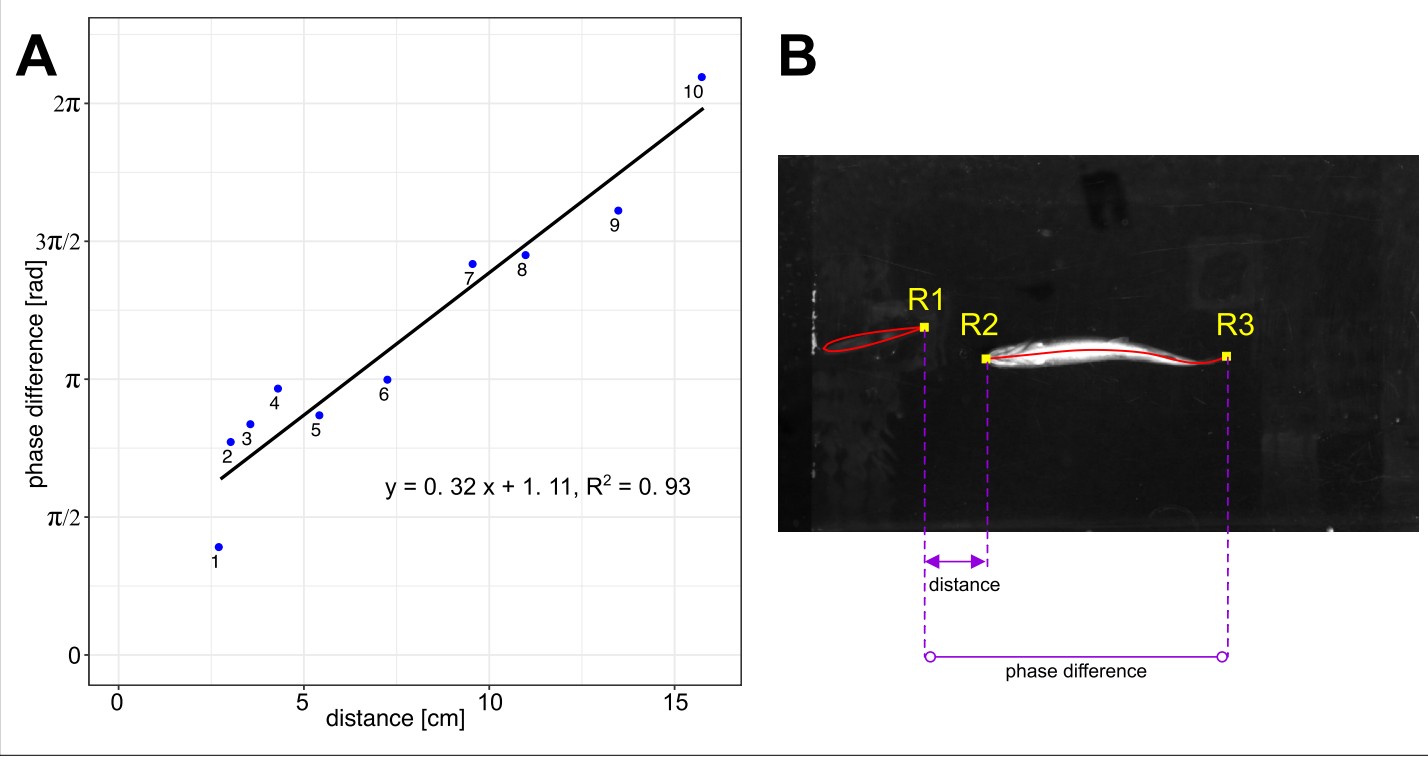

**Figure 4.** Phase difference between foil and fish. (**A**) Linear relationship (n = 10) between phase difference and distance from the foil for fish swimming in-line in the thrust wake. *Video 3* shows videos of the individual data points 1–10. (**B**) Distance is measured between the trailing edge of the foil (R1) and the leading edge of the fish (R2). The phase difference is measured between trailing edges of the foil (R1) and the fish (R3).

their position in the thrust wake by beating their tails less often than when they swam at the same ground speed in free-stream flow. These experiments further showed that fish were synchronizing their tail-beat frequency to that of the flapping foil. For both smaller (mean ± s.d.: 2.15 ± 0.29 Hz) and larger (1.96 ± 0.04 Hz) fish the swimming frequency approached the 2 Hz flapping foil motion when swimming in the foil thrust wake.

We analyzed body bending kinematics and identified decreased overall phase lags along the body in the thrust wake in both size groups (*Figure 3B*) compared to the free stream control condition. As a result, the bending of consecutive body segments was timed closer together (*Figure 3C*). Smaller overall phase lags also relate to fewer waves along the body. We did not find any significant differences in body amplitude between fish that swam in thrust wakes and in the free stream (*Figure 3— figure supplement 1*).

Reduced tail-beat frequencies towards the ones of the flapping foil and a change in body phase lags indicate that fish are synchronizing their movements to the flapping foil. To further investigate synchronization, we measured the phase difference between fish and the flapping foil as a function of the distance between them (*Figure 4*, *Video 3*). We found a linear relationship ($R^2 = 0.93$) showing that the phase difference increases as fish are swimming further away from the foil. This result demonstrates that fish time their body undulations and tail-beats depending on their location in the thrust wake, and it suggests that they synchronize their movements to the oncoming vortices that are shed by the flapping foil.

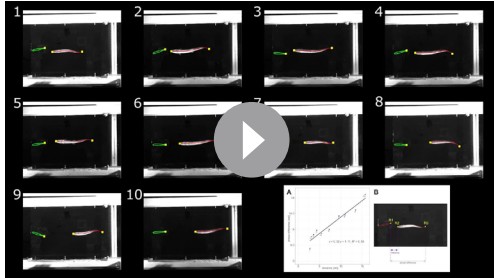

**Video 3.** Analysis of phase difference between flapping foil and fish swimming in the thrust wake. Panels 1-10 show the individual data points and illustrate how the phasing of the tail-beat changes linearly with the distance from the foil. Numbers on the panels correspond to the point numbers in the graph in the lower right, and to those in *Figure 3*.

https://elifesciences.org/articles/81392/figures#video3

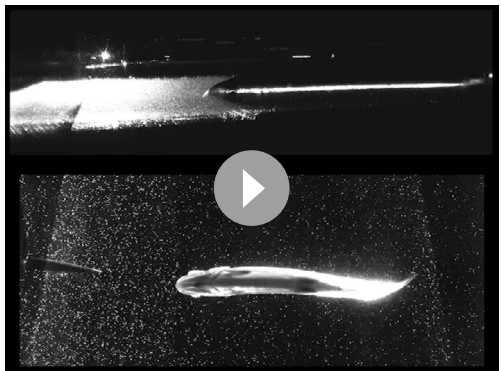

**Video 4.** Trout swimming in a laser sheet used for particle image velocimetry (bottom and side views). https://elifesciences.org/articles/81392/figures#video4

## How do fish interact with the flow in thrust wakes?

Kinematic analysis of fish swimming in thrust wakes indicates a frequency and phase synchronization with the flapping foil. To investigate flow dynamics and how the thrust wake generated by the foil interacts with the bending fish body we employed particle image velocimetry ('Materials and methods: Setup to capture flow dynamics,' *Video 4*) to visualize flow structures in the thrust wake during fish swimming trials. Analysis of flapper wake velocity fields show that trout in the thrust wakes interact with oncoming vortices that are shed from the flapping foil and time their movements accordingly. We identified two scenarios that we call double-sided and single-sided vortex interaction. Double-sided vortex interaction (*Figure 5A–F*, *Video 5*) is characterized by an initial vortex interception which splits the vortex in two parts. One part of the vortex stays attached and 'rolls'' downstream along the body, whereas the other part is shed laterally and moves away from the body. In this situation, the trout body alternates between clockwise and counter-clockwise vortices that are intercepted, stay attached and roll along corresponding alternate sides of the body. Fish are thus able to 'catch vortices on both sides of the body.

Single-sided vortex interactions (*Figure 5G–L*) undergo the same process of 'catching vortices'; however, only for one of the two differently oriented vortex types that are shed from the foil. Consequently, the intercepted vortex stays attached and 'rolls' only along one side of the body. The single-sided vortex interactions are related to a slight lateral offset of fish position with respect to the center line around which the foil is oscillating, whereas double-sided vortex interactions occur for fish swimming directly in the center line. We also note that vortex interactions occurred at different distances with respect to the foil (d1 and d2 in *Figure 5*). The consistent interaction pattern between the trout body and oncoming vortices indicates that these fish are synchronizing their movements with respect to the flapping foil and the corresponding vortices shed into the wake.

## Decreased head pressure indicates reduced energy requirements

A large part of the drag on a swimming fish at Reynolds numbers greater than 5000 is caused by drag forces at the anterior portion of the body that faces the oncoming flow (*Du Clos et al., 2019*; *Lucas et al., 2020*). Total body pressure drag on a swimming streamlined fish like trout is mainly determined by the pressure acting on the head ($F = p \cdot S$, F: drag force, p: pressure, S: surface area). Therefore, to estimate the effect of swimming in a thrust wake on drag we compared pressure fields of fish swimming in the free stream to thrust wake conditions.

We derived the pressure fields at the anterior part of the fish body based on velocity field changes as proposed by *Dabiri et al., 2014* ('Materials and methods: Pressure field computation'). The computed pressure fields revealed reduced average head pressures in the thrust wake (*Figure 6A and B*, *Figure 6—figure supplement 1*). We found the strongest decreases (46% and 86% decrease compared to free-stream swimming) for fish swimming close to the foil and exploiting double-sided vortex interactions. Fish swimming further away from the foil and exhibiting single-sided vortex interactions also showed reduced average head pressure magnitudes compared to free-stream swimming (45% decrease). Here, we found an asymmetric average pressure pattern with higher average pressures at the side closer to the centerline of the foil oscillation. The other side of the head experienced smaller average pressures.

To understand how the average head pressures in the thrust wake were reduced despite faster oncoming flows caused by the flapping foil, we analyzed the instantaneous pressure fields (*Figure 6D1–D4*). Here, it becomes evident that the flapping foil induces oscillating negative and positive pressure zones around the head. The negative pressure (suction) zones cause forward thrust forces, whereas the positive pressures contribute to drag. On average, this reduces overall head drag

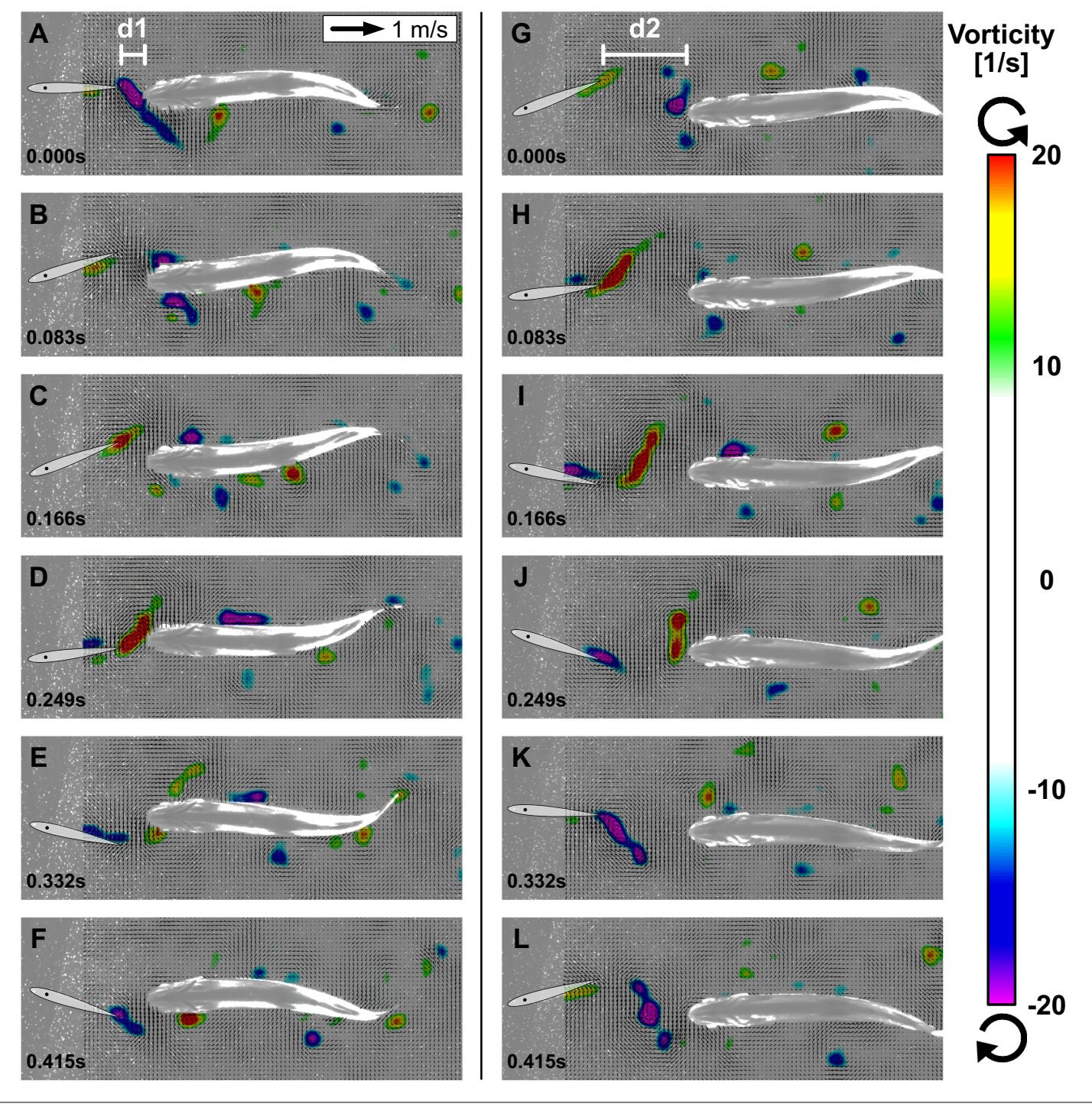

**Figure 5.** Interactions between fish and vortices. Two representative sequences over one swimming cycle with ventral view of trout station-holding in the thrust wake near the foil at distance d1 with double-sided vortex interactions (**A–F**) and located more downstream at d2 with single-sided vortex interactions (**G–L**). Oncoming vortices from the flapping foil are intercepted by trout in the wake. The vortices stay attached on one side depending on their orientation and 'roll' downstream along the body (velocity fields shown after subtraction of mean flow speed).

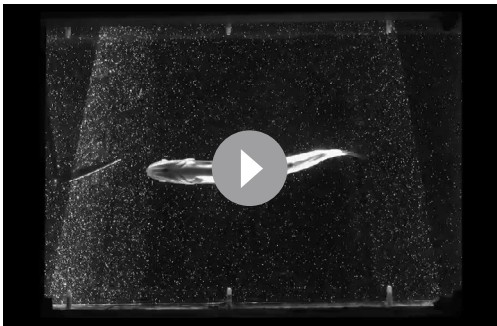

**Video 5.** Visualization of vortex flow structures that interact with a trout swimming in the thrust wake (bottom view).

https://elifesciences.org/articles/81392/figures#video5

as the positive pressure magnitudes are comparable to free-stream swimming but mean head pressure is reduced by the occurrence of negative head pressures for part of the cycle. This pressure analysis indicates that the drag of fish swimming in thrust wakes is reduced compared to free-stream swimming, and therefore supports the hypothesis of decreased energy used to hold station in thrust wakes with accelerated mean flow.

## Discussion

In schools of swimming fishes, there are a number of different hydrodynamic effects that can be exploited to save energy by individual fish in various positions (*Figure 1*). Previous work has demonstrated benefits for swimming side-by-side (phalanx configuration), pushing off near followers, and forming diamond patterns (*Ashraf et al., 2017*; *Saadat et al., 2021*; *Taguchi and Liao, 2011*). But fish in schools often assume an in-line configuration with one fish swimming directly behind another (*Video 6*). The benefits, if any, of swimming in this

**Figure 6.** Reduced head pressure in the thrust wake. Average pressure fields of a trout swimming in free-stream flow (**A**) and in the thrust wake of a flapping foil (**B**) show reduced positive pressures (46% decrease) around the head despite increased oncoming flow. Consistent instantaneous positive pressures over time are present under free-stream flow conditions (**C1–C4**). Corresponding instantaneous pressure fields display alternating positive and negative pressures around the head in the thrust wake over time (**D1–D4**).

The online version of this article includes the following figure supplement(s) for figure 6:

**Figure supplement 1.** Reduced average head pressures in thrust wakes.

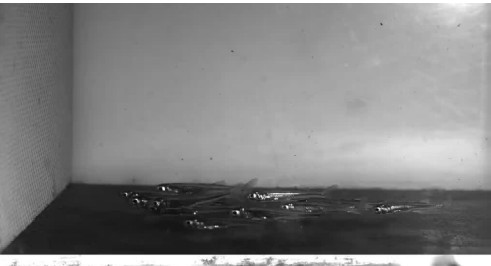

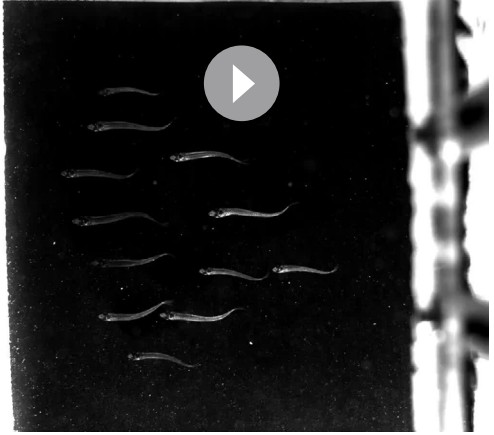

**Video 6.** Schooling silversides, *Menidia menidia*, swimming in a flow tank and exhibiting in-line swimming (bottom and side views).

https://elifesciences.org/articles/81392/figures#video6

tandem swimming mode have been the subject of some debate. Some authors (*Verma et al., 2018*; *Weihs, 1973*) suggest that swimming in tandem is not an energetically favorable configuration due to the accelerated wake flows generated by the fish in front. Other, primarily computational studies have suggested that a trailing streamlined shape could in fact experience reduced energetic cost due to leading edge suction resulting from an oscillating flow impinging on the head or leading edge of the trailing fish or foil (*Kurt and Moored, 2018*; *Maertens et al., 2017*; *Saadat et al., 2021*). To date, however, no experimental study has demonstrated that live fish will voluntarily swim in a thrust wake and that reduced swimming cost could result from such a position. With these experiments, we document that trout indeed perform volitional in-line swimming with their body located within the accelerated flow region, and our analysis suggests that they can save energy under these conditions.

## Comparisons to drag wake swimming and differences from drafting

A drag wake in the context of fish swimming and schooling is characterized by a von Kármán vortex street between two thrust wakes (e.g., shed by two fish swimming parallel to each other). Drag wakes can be emulated behind cylinders when they are exposed to sufficiently high flow speeds. It is important to note that the average flow speed in a drag wake is inherently slower than in the free stream. This is highlighted in experiments that demonstrate a dead fish propelling itself forward in the wake of a cylinder (*Beal et al., 2006*). Intuitively, we can draw an analogy of a cyclist drafting behind another cyclist where the individual behind experiences reduced energy consumption while maintaining the same speed. This situation is an example of a drag wake and the reduced costs that ensue from moving in that reduced velocity zone: trailing cyclists benefit from the reduced relative oncoming flow which results in reduced aerodynamic drag.

The dynamics of thrust wakes, however, differ from drag wakes. Vortex orientations are reversed compared to the drag wake (termed a reverse Kármán vortex street), and, notably, thrust wakes are characterized by a higher average flow speed than in the free stream. For swimming fish, this is a consequence of tail fin movement which actively generates thrust so that individual fish located behind this thrust wake experiences higher than free stream mean flow velocities. Given this increased oncoming flow speed it is surprising that fish choose to swim in thrust wakes. If we return to our example of cyclists, it would correspond to a (fictional) case where a leading cyclist would have a propeller attached to their bicycle that generates additional thrust. The trailing cyclist would face an increased oncoming flow and experience increased aerodynamic drag. An in-line, tandem, formation might be expected to be disadvantageous in this case.

How does this situation differ from fish swimming in a thrust wake of a conspecific? A key difference is the undulatory characteristic of thrust wakes that are produced by swimming fish. A trailing fish faces the oncoming flow at its head with an oscillating angle of attack, and (unlike the trailing cyclist) the trailing fish oscillates its head during swimming (*Di Santo et al., 2021*) further enhancing the time-dependent variation in flow in the head region. Our analysis showed that as a result of the oscillatory wake impinging on the fish head the pressure distribution in the head region is composed of both positive and negative pressures, and thus effectively reduces the overall pressure drag. This

is in agreement with previous simulation analyses (*Maertens et al., 2017*; *Saadat et al., 2021*) and makes in-line swimming an advantageous formation for fish.

## Swimming efficiency in thrust wakes

In our experiments, we found that fish in thrust wakes significantly reduced their tail-beat frequency and the frequency, higher when swimming in the free stream, shifted toward the flapping foil frequency. We also found a linear relationship between phase difference and distance from the foil when fish swam in-line in the thrust wake, indicating that they are phase synchronized with the foil. Together with our PIV analysis, these results suggest that trout are exhibiting vortex phase matching (*Li et al., 2020*) which has been identified as an energy saving mechanism. These benefits are further supported by our computational analysis where we found lower pressure drag in fish swimming in the thrust wake. Apart from the reduced hydrodynamic resistance, the lower tail-beat frequencies without significant changes in amplitude also are reflective of reduced metabolic cost (*Ohlberger et al., 2007*; *Steinhausen et al., 2005*). The change in phase lags that we observed further suggests a change in the muscle activation pattern along the body. Decreased muscle activation was observed by *Liao et al., 2003* when trout swim with a Karman gait in a drag wake, and passive dynamics could therefore be additional sources for energy savings that would however need to be confirmed in future experiments. Overall, our data suggests that fish can swim more efficiently in thrust wakes because they maintain the same swimming speed as in the free stream while facing reduced pressure drag and spending less energy by beating their tail less frequently.

These results are in agreement with past simulation and robot studies. (*Harvey et al., 2022*) adopted a similar approach to ours and exposed rainbow trout to a hydrofoil thrust wake. They also observed altered swimming gaits and estimated energy savings but via measurements of acceleration. It should also be noted that they used juveniles about three times smaller than trout used in our study, however at similar foil cord length. This indicates that fish in thrust wakes may exhibit energetic benefits over an extended size scale. (*Verma et al., 2018*) explored simulated leader-follower formations in a reinforcement learning framework. A first set of optimizations with a reward function based on a modified Froude efficiency led to formations in which followers *settled close to the center line of the leader's wake* and showed well-coordinated behavior of the follower with the wake. In the same work, *Verma et al., 2018* concluded in a second set of simulations that swimming in-line with a leader is not associated with energetic benefits for the follower. It is important to note that this conclusion was drawn based on a swimming strategy in which the follower would strictly try to attain an in-line position regardless of energetic considerations. As a consequence, the optimization led to an increased swimming amplitude, which permitted in-line swimming but at a higher energetic cost. We confirm that in-line swimming by itself is not necessarily energetically beneficial. More importantly, efficient swimming requires the correct timing of interactions with the wake. Rather than viewing in-line swimming as a policy, we can see in-line positions as favorable conditions to maintain wake synchronization as we found in our experiments of double-sided vortex interactions.

Finally, in-line swimming has been dismissed as a beneficial strategy in the past considering the diverging characteristics of three-dimensional compared versus two-dimensional wakes (*Verma et al., 2018*). In such cases, it could be argued that the area around the centerline of the wake is composed of quiescent flow and in-line swimming offers no opportunity to interact with vortices. Whereas these diverging wakes are predominantly found in simulation studies at lower Reynolds numbers (*Borazjani and Sotiropoulos, 2008*; *Liu and Dong, 2016*; *Verma et al., 2018*), we found no evidence for bifurcating wake structures behind trout swimming in the free stream, a finding in line with previous analyses of trout wake flow patterns (*Müller et al., 1997*; *Tytell, 2010*). The artificial thrust wake generated using our robotic flapping foil produced a parallel vortex street similar to our observations of the wake in freely swimming trout. In addition, given the experimental data on single-sided vortex interactions, we hypothesize that energy-efficient thrust wake interactions could also occur in diverging wakes but with a small offset to the centerline.

## Limitations of this study

Previous studies have highlighted the three-dimensional (3D) effects of fish swimming. The 3D kinematics of the tail are determined by the 3D body shape as well as the motion (*Tytell et al., 2008*), and it has been shown that the tips of the caudal fin are subject to cupping into the flow. The flapping foil

in our study is rigid and does therefore not exactly replicate this motion pattern. Future work could address this gap by using a flexible flapping foil. Another aspect of 3D fish swimming are 3D vortex rings that are shed into the thrust wake at the caudal fin. These structures induce flow in (1) lateral (side-to-side) and axial (forward-backward) swimming directions as well as in (2) the vertical (up-down) swimming direction. The flapping foil in our experiments spanned across an extended depth ('Materials and methods,' *Figure 2*) and generates 2D thrust wakes that produce the lateral and axial flow characteristics of vortex rings. These flow components are arguably important for thrust generation to swim forward and we showed in our study how fish interact with these flows and benefit from reduced pressure drag. Vertical components need to be included in future experiments to address how the up-and-down flow dynamics impact fish in a thrust wake. Nonetheless, our results contribute to a better understanding of in-line swimming in thrust wakes and are likely to extrapolate to 3D fish swimming.

In this study, we analyzed a limited number (n = 3) of swimming trials using PIV. Our goal was to use these trials to investigate the underlying mechanism of vortex interactions following the kinematic analysis that showed both frequency and phase synchronization between fish and the flapping foil. We were able to gain insight on single and double-sided vortex interactions and on reduced pressure drag via computational inference. We expect that there is a critical distance from the foil centerline at which fish transition from double to single-sided interactions. To further identify and quantify this critical distance a larger sample size is required. We will also require more swimming trials to quantify the reduction in pressure drag depending on the distance from the foil. This could help to understand if and where there are distances that are optimal in terms of energy savings, as, e.g., suggested in *Saadat et al., 2021*.

We know from past studies that there are a number of hydrodynamically beneficial schooling positions (*Figure 1*). Our results complement this body of work with regard to fish swimming in thrust wakes that are shed by leading individuals, a condition encountered within fish schools during in-line locomotion. In our controlled experiments, we show that trout volitionally swim in thrust wakes and exhibit advantageous flow interactions with incoming vortices suggesting increased energy efficiency for the in-line swimming condition. These results highlight the hydrodynamic complexity of fish schooling and support a view in which individuals in schools have a variety of opportunities to save energy when they swim side-by-side, in drag wakes, and in thrust wakes behind each other.

## Materials and methods

### Animals
We used brook trout, *S. fontinalis,* and carried out the same experiments 2.5 months apart to investigate size effects as the total lengths of the fish increased after this growth period. Smaller fish had a body length of BL = 15.8 ± 0.5 cm (n = 4, Re = 43,091, Re = $\frac{u \cdot BL}{\nu}$, u=0.3 $\frac{m}{s}$, $\nu = 1.1 \cdot 10^{-6} \frac{m^2}{s}$) and larger fish had a body length of BL = 19.3 ± 1.0 cm (n = 6, Re = 52,636). Particle image velocity trials were carried out for larger fish (n=3). Trout were held at a water temperature of 16°C and all experiments were performed in accordance with Harvard animal care and use guidelines, IACUC protocol number 20-03-3 to GL.

### Experimental setup
We carried out all our experiments in a custom flow tank with flow speed control (*Figure 2*, *Video 1*). Fish were able to move freely in a space of 28 cm × 28 cm × 64 cm, where the front and back portions of the swimming section were limited by baffles. Given fish body widths of 3 cm or less, the side-to-side dimensional comparison of tank width to trout width has a ratio of about 10:1. Corrections for fish blocking of flow are not needed for less than 10% of the cross-sectional area (*Kline et al., 2015*). The boundary layer thickness in this tank is approximately 5 mm and has been quantified in *Tytell and Lauder, 2004*. We also only considered swimming trials in the center region of the tank for our analysis.

### Flapping foil
We used a symmetric 3D printed NACA 0012 airfoil (cord: 67 mm; span: 190 mm; thickness: 8.1 mm; center of rotation: 48 mm from trailing edge; material: transparent photopolymer [RGD810] from an

Connex 500 3D printer) as in previous studies of biomimetic propulsion and to emulate a fish body shape (*Karbasian and Esfahani, 2017*; *Lauder et al., 2007*; *Shua et al., 2007*; *Van Buren et al., 2019*; *Zhang et al., 2019*). It was actuated in sway and yaw direction (*Figure 2*) to mimic the tail-fin portion of swimming fish to induce fish-like thrust wakes. It is worth noting that we are using a rigid foil whereas the fish tail is flexible. Our goal was to reproduce similar tail tip excursions, therefore the corresponding movement was parametrized as follows:

$$y_{sway} = a_{sway} sin\left(2\pi ft\right), \tag{1}$$

$$\phi_{yaw} = a_{yaw} sin\left(2\pi ft - \frac{\pi}{2}\right) \tag{2}$$

Here, $a_{sway}$ and $a_{yaw}$ denote the sway and yaw amplitudes, respectively. $f$ indicates the flapping frequency, and $t$ the time in seconds. The two motions are offset by a phase shift of $\frac{\pi}{2}$, which ensures that maximal yaw is reached whenever the foil crosses zero sway. For the purpose of our experiments $a_{sway} = 1$ cm, $a_{yaw} = 20°$ were selected and resulted in a peak-to-peak tail-beat amplitude of $A = 4$ cm, which is comparable to the width of wakes in fish. Together with a frequency of $f = 2$ Hz, this resulted in a Strouhal number of $St = \frac{A \cdot f}{U} = 0.267$ and a Reynolds number of $Re = \frac{UL}{\nu} = 20100$ with $U = 0.3 \frac{m}{s}$, $L = 6.7$ cm (cord length), and $\nu = 10^{-6} \frac{m^2}{s}$, thus operating in a turbulent flow regime.

## Use of NACA 0012 foil

A large body of work suggests that NACA 0012 foils are appropriate for the purpose of creating fish-like wakes (*Anderson et al., 1998*; *Triantafyllou et al., 1993*; *Triantafyllou et al., 2004*). They demonstrate that these foils show high propulsive efficiencies if they are operated in a Strouhal number range of 0.25–0.35 and that the reverse Karman vortex street with its increased wake velocities is related to thrust production. In part, high efficiency is associated with leading-edge vortices that are convected downstream and interact with trailing-edge vortices that result in a reverse Karman street (*Anderson et al., 1998*). Furthermore, previous studies have shown that fish operate in a similar range of Strouhal numbers and also produce reverse Karman vortex streets (*Saadat et al., 2017*). In contrast to these studies, classic low Reynolds number (Re < 100,000) airfoil literature suggests that NACA 0012, a supercritical airfoil, suffers from laminar separation bubbles and high drag when operated at lower Reynolds numbers. This could indicate that there are more efficient foils, e.g., cambered, flat or flexible foils, that could be used to generate the wakes as in our study. These foils are more suited for subcritical Reynolds numbers and future studies could explore differences and similarities by comparing sub- and supercritical foils in the context of creating fish-like wakes. However, as mentioned above, we were able to generate thrust producing wakes characterized by reverse Karman vortex streets with increased wake velocities that were sufficient for the purpose of our experiments as they provide the flow structures that fish face in wakes of other fish.

## Wall effects

We have previously investigated wall effects (*Quinn et al., 2014a*) for the same flow tank and with the same robotic flapper used in this study. Using a six-axis force/torque sensor propulsion speed, forces, and efficiency were quantified at varying distances from the wall. The results of this analysis show that no significant wall effects are to be expected in our experimental setup of trout swimming behind a flapping foil. In brief, *Quinn et al., 2014a* studied the propulsion of relatively large flexible panels (150 mm span by 195 mm length) at varying distances from both the wall and bottom of the flow tank. The panels, when moved, came as close as 15 mm from the wall, which is closer than the trout studied here that were in the middle of the tank (~30 cm on a side and 1 meter long working area for filming). Plastic panels of three flexural stiffnesses were studied (*Quinn et al., 2014a*; table 1) and these panels were moved at the leading edge (in a manner similar to the foil motion used in this manuscript) to generate a propulsive wave. The flexural stiffness of the panels in *Quinn et al., 2014a* was specifically chosen to encompass the range of actual fish flexural stiffness which range from 10 to 3 to 10–6 (*Shelton et al., 2014*) and panel B (*Quinn et al., 2014a*; table 1) matches the flexural stiffness of trout bodies.

Given the non-dimensional distance d/a (d = mean distance from the wall, and a = heave amplitude of the flexible body at the leading edge), *Quinn et al., 2014a* show that there is little to no effect on propulsive speed and economy for all panels at d/a > 5. For the most trout-like panel B, there is almost

no effect at any distance. In this article, trout operated at a value of d/a between 7 and 3.5 depending on whether 'a' is taken as head oscillation or tail oscillation amplitude.

One additional study from *Quinn et al., 2014b* used another flapper/flow tank system to investigate the wall effect for a rigid pitching foil at different distances from the wall. Their data show that for conditions similar to that of our trout experiments where trout are 15 cm from the wall, there is no effect on propulsive forces.

Although swimming close to a surface with an undulatory body can certainly improve propulsive efficiency and alter the time-dependent profiles of forces and torques, these previous experiments using the exact same experimental system show that it is highly unlikely that wall effects have influenced our results for trout swimming in the center of the flow tank.

## Fish kinematics

High-speed video was used to capture kinematic variables such as tail-beat frequency, body amplitudes, and phase lags during swimming trials. The experiments were carried out in the dark to provide a controlled environment with minimal external distractions. To provide the fish with some sense of visual orientation, a fiber light was installed upstream behind the front baffle in the flow tank. We then used infrared lights (*Figure 2*), which are outside the visual spectrum of trout, to provide the necessary illumination to capture the scene with high-speed cameras. We took video recordings at 125 frames per second from a ventral and an angled side view.

## Particle image velocimetry

To capture the flow patterns during swimming trials we used particle image velocimetry (PIV) as in our previous work (*Domel et al., 2018*; *Thandiackal et al., 2021b*; *Zhu et al., 2019*). For this purpose, we seeded the water in the flow tank with near-neutrally buoyant plastic particles (~50 μm mean diameter) and used two lasers to create a light sheet around the swimming fish (*Figure 2*). Movements were then recorded at 1000 frames per second from a ventral and angled side view (*Video 4*). We used the side view to identify the location of fish with respect to the light sheet. Only swimming sequences where the laser light sheet passed through the middle of the swimming fish body were considered in our analysis.

## Kinematic analysis

### Body midlines

We used a custom MATLAB script to manually track 9 points along the body midline within a given frame. Piecewise cubic spline interpolation was then applied to generate smooth midline curves. We manually tracked the midlines in every sixth frame and linearly interpolated between these frames to obtain midline estimates for all frames that were recorded at 125 frames per second.

### Frequency estimation

Tail-beat frequencies were determined by averaging the period between maximal lateral tail tip excursions over three consecutive swimming cycles for each swimming trial.

### Phase lag estimation

The phase lag describes how the traveling wave of body bending propagates along the body. To quantify the body bending, we divided the body first into $N = 20$ equal length segments and computed the joint angles $\phi_i$ between segments. The intersegmental phase lag $\Delta\phi_i$ was then computed as the time delay between joint angles of consecutive segments as a fraction of the cycle duration $T$. As in *Thandiackal et al., 2021a*, we used cross-correlation of the joint angle signals to determine this time delay. Finally, we obtained an estimate of the overall phase lag $\Delta\Phi$ by summing up the intersegmental phase lags along the entire body.

$$\Delta\phi_i\left(t\right) = \frac{crosscor\left(\phi_i,\ \phi_{i+1}\right)}{T} \cdot 2\pi, \tag{3}$$

$$\Delta\Phi\left(t\right) = \sum_{i=1}^{N} \Delta\phi_i\left(t\right) \tag{4}$$

We note that the wavelength $\lambda = \frac{2\pi}{\Delta\Phi}$ (as a fraction of the body length) can be computed from the phase lag, and we report this metric in the supplementary data (*Thandiackal and Lauder, 2022*).

### Body amplitude estimation

We define the amplitude along the body as the maximal displacement perpendicular to the forward direction of movement. Forward and lateral direction are determined by applying a principal component analysis on the point cloud of all tracked midline points from a given swimming trial. The first principle component (PC) that captures most of the variation represents the forward direction whereas the second PC represents the lateral direction. Based on these directions the lateral displacement at a given midline point in time is then determined as the projection of that point on the lateral direction. Finally, we determine the body amplitude as half of the range of lateral displacements at a given midline point over the duration of the swimming trial.

### Phase difference estimation

To estimate the phase difference between the flapping foil and fish swimming in the thrust wake, we compared the lateral displacement of the trailing edges of the foil and fish. As for the body phase lag (see above), we used a cross-correlation of these two signals to determine the time delay as a fraction of the cycle duration.

### Statistical analysis

To confirm hypothesized decreases in mean frequency and phase lag between the free stream control condition and swimming in the thrust wake, we carried out paired, one-sided Welch *t*-tests (assuming unequal variance). Significant differences in mean amplitude under these conditions were investigated using paired, two-sided Welch *t*-tests. p-Values are reported in *Figures 2 and 3*.

### Pressure field computation

Pressure fields were inferred from particle image velocimetry (see 'Experimental setup'). We followed the methodology described in our previous work (*Thandiackal and Lauder, 2020*). In brief, the pressure at each grid point is computed by taking the median over eight families of integration paths that each integrate the pressure gradients. The pressure gradients themselves are estimated based on sequential velocity fields and zero pressures are assumed at the domain boundary where paths are initiated. Velocity fields were computed in DaVis 8.3 (LaVision Inc) and the pressure fields were obtained using the Queen 2.0 software by *Dabiri et al., 2014*. Corresponding fluid–solid interfaces, that block integration paths, included both the flapping foil as well as the fish body and were extracted using custom MATLAB and Python scripts. This approach has been used and validated in previous publications (*Lucas et al., 2017*; *Lucas et al., 2020*; *Thandiackal et al., 2021b*).

We quantified the head pressures by averaging over a rectangular zone that extends from the fish snout and that spans 10% of the body length in axial direction and the width of the fish in lateral direction. Albeit arbitrarily defined, this allowed us to directly compare pressure drags for steady swimming vs. thrust wake swimming.

## Acknowledgements

This work was supported by the National Science Foundation (Grant number EFRI-830881) and the Office of Naval Research (Grants N00014-18-1-2673, N00014-14-1-0533, and N00014-21-1-2210). We thank members of the Lauder Lab for many helpful discussions about in-line swimming, and Prof. Valentina DiSanto for collaborative research on *Menidia* schooling shown in Video 6. Publication charges paid by a grant from the Wetmore Colles fund, Museum of Comparative Zoology, Harvard University.

## Additional information

### Funding

| Funder | Grant reference number | Author |
|---|---|---|
| National Science Foundation | EFRI-830881 | George Lauder |
| Office of Naval Research | N00014-18-1-2673 | George Lauder |
| Office of Naval Research | N00014-14-1-0533 | George Lauder |
| Office of Naval Research | N00014-21-1-2210 | George Lauder |

The funders had no role in study design, data collection and interpretation, or the decision to submit the work for publication.

### Author contributions
Robin Thandiackal, Conceptualization, Data curation, Software, Formal analysis, Investigation, Visualization, Methodology, Writing - original draft, Project administration, Writing - review and editing; George Lauder, Conceptualization, Supervision, Funding acquisition, Methodology, Writing - review and editing

### Author ORCIDs
Robin Thandiackal ⓘ http://orcid.org/0000-0001-8201-4892
George Lauder ⓘ http://orcid.org/0000-0003-0731-286X

### Ethics
All experiments were performed in accordance with Harvard animal care and use guidelines, IACUC protocol number 20-03-3 to George V. Lauder.

### Decision letter and Author response
Decision letter https://doi.org/10.7554/eLife.81392.sa1
Author response https://doi.org/10.7554/eLife.81392.sa2

## Additional files

### Supplementary files
• MDAR checklist

### Data availability
Data that support the findings of this study are available on: https://doi.org/10.6084/m9.figshare.c.6093405; https://gitlab.com/robintha/052-matlab-midline-annotation (copy archived at swh:1:rev:f3e2d4b0afb18419ac816b7df45f5ad0630de59c); https://github.com/basilisklizard/fluid-structure-segmentation (copy archived at swh:1:rev:7383c1d1627b2285b6a29074479f78220e3bd502).

The following dataset was generated:

| Author(s) | Year | Dataset title | Dataset URL | Database and Identifier |
|---|---|---|---|---|
| Thandiackal R, Lauder GV | 2022 | Data and Movies - In-line swimming dynamics revealed by fish interacting with a robotic mechanism | https://doi.org/10.6084/m9.figshare.c.6093405 | figshare, 10.6084/m9.figshare.c.6093405 |

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
