## [Editor Report]

Why do fish school together? Energetic benefits have long been considered a key factor in motivating fish to swim together and tune their tailbeat to exploit the whirling wake generated by conspecifics. This study clearly demonstrates that fish benefit from swimming in a two-dimensional vortical wake by locating their body in the vortical low-pressure zones that passively impart a net thrust force on their oscillating bodies. The behavioural and biofluid mechanical findings will interest comparative biomechanists, movement ecologists, evolutionary biologists, fluid mechanists, and bioinspired roboticists.

---

## [Decision Letter]

**Decision letter after peer review:**

Thank you for submitting your article "In-line swimming dynamics revealed by fish interacting with a robotic mechanism" for consideration by *eLife*.

Your article has been reviewed by two peer reviewers, and the evaluation has been overseen by David Lentink as the Reviewing Editor and Christian Rutz as the Senior Editor. The following individual involved in the review of your submission has agreed to reveal their identity: Maurizio Porfiri (Reviewer #2).

The reviewers have discussed their reviews with one another, and the Reviewing Editor has drafted this decision letter to help you prepare a revised submission.

Essential revisions:

1) In the summary of potential energy-saving mechanisms in schooling fish, please include the "vortex-phase matching" mechanism (ref. 33 in the references); it is related to the mechanism found in this paper, as discussed in the Discussion section (Line 291).

2) The justification for the effectiveness of the rigid NACA airfoil is not clear. In Line 108, you mentioned "similar hydrodynamic characteristics", but what are these characteristics? In Line 125 and the Materials and methods, you mentioned similar reverse Karman vortex streets and increased speeds in the wake. A quantitative comparison would help justify and quantify the similarity. Further, we are missing a discussion of the Reynolds number regime. NACA airfoils are super-critical airfoils, meaning that they have been designed for high Reynolds numbers during which laminar flow separation is a minor issue. Operating such super-critical airfoils at sub-critical Reynolds numbers usually results in large-scale flow separation, such as laminar separation bubbles, poor stall behavior, and a drag crisis. Variable camber and a 3D shape (a fish-like body) or sharp leading edge of a 2D airfoil (optimal sub-critical airfoil shape) can dramatically reduce the issues. This is well established in the low Reynolds number aerodynamics literature and applies 1:1 to hydrodynamics, but is often overlooked by biomechanists working on swimming. Please outline the issues as limitations of the study in the methods and Discussion sections.

3) The novelty of the robotics-based experimental approach is overstated, given that several earlier studies have investigated the response of live fish to thrust wakes, generated by pitching airfoils or robotic fish. Please give due credit to previous studies and temper claims accordingly. Such strong novelty claims are not needed to make a compelling case for publication in *eLife* -- our policy is to focus on informative scientific contributions in the context of the literature.

4) It seems like the length of the test section in the experiments is on par with the body length of the animals, hence wall-effects could confound the study to some degree. Please use the established literature and any estimation within your reach to outline wall-effects in the methods, and discuss the results in light of this limitation. Provide guidance for follow-up studies on how they could improve on the current experimental design.

5) The role of 3D effects needs to be elaborated; the 2D experimental data limit the generality and conclusions on actual fish collective behavior. Please include these issues in the introduction, explain how you converged on this experimental design in the methods, and discuss the limitations in the discussion and how future studies could expand the current work to the full 3D realm. In the discussion, please cite papers comparing 2D and 3D wakes to infer any possible implications for interpreting the quantitative measurements of vorticity and pressure in the paper, and extrapolating these findings to 3D fish swimming close together. The merit of 2D studies is not questioned, but does limit the conclusions and should inspire future 3D studies to further test your conclusions and extrapolate the implications for actual (3D) fish collective behavior.

6) Please briefly comment on other sensory modalities (touch, the vestibular system, vision), citing the relevant literature, before going in-depth on which senses play a role, so general readers (across biology and engineering) can understand your framework and its justification/limitations. Expanding the introduction and discussion on sensory modalities beyond the chosen/motivated focus will help readers understand your scientific reasoning.

7) In the methods section, please provide detailed reasoning for the selection of particle image velocimetry plane chosen for the computation of vorticity from 2D data and inference of pressure from 2D data. Although the cited paper from Dabiri is helpful, a summary of the approach for pressure inference is essential to clarify its reliability. This knowledge cannot be assumed to be generally known among the readership.

8) Please clarify if the flow is laminar or turbulent in the tested flow regime and add a plot showing at what distance the switch between double vortex split and single-sided vortex occurs.

9) Please carefully double-check the accuracy of wording in "increase in flow speed in the thrust wake" in the abstract and introduction, which may be in contrast with flow refuging behind bluff bodies.

10) In Figure S2, it appears that the fish and airfoil differ, yet they appear to shed similar vortex pairs -- is this the case? If so, please discuss this matter in your experimental design (and discussion where relevant).

11) Line 158: The "one-sided" versus "two-sided" vortex interaction analyses are qualitative. Is it possible to quantitatively analyze the relationship between the phase of the airfoil and the phase of head movements, since the thrust wake is linked to the phase of the airfoil? In Line 184, you mentioned fish adopt two-sided or one-sided interactions depending on the lateral offset and the distance between the airfoil and real fish. Is it possible to estimate the critical distances which drive the fish to switch from two-side vortex interaction to one-side?

12) Since fish might also swim further away (e.g., S Movie 2), what are the parameters you used to define/filter in-line swimming patterns? Line 165: You state "time their movements accordingly" to interact with the wakes generated by the airfoil -- but is it possible to add quantitative analyses of how the fish may time their movements? For example, by determining the phase difference between the airfoil motion and fish head oscillation? This seems also applicable to Line 142. Finally, in Line 146, please explain to the general reader why you did not consider using fish body wavelength for determining phase lag, which is more typical for fish kinematics. If this is not possible, please justify your processing choices accordingly, so the general reader can understand your choices and any limitations this may have for interpreting the findings.

13) Line 122 states you chose St around 0.267. Please outline how you selected the oscillation frequency of the airfoil with respect to the natural tail beat frequency. Also, please clarify how you selected the experiment parameters a_{sway}, a_{yaw}, amplitude, since there are different combinations of these parameters possible that give St ~ 0.267, and not all combinations are biologically representative or informative. Please include typical kinematics data of real fish here to help justify the parameter selection and, whenever there is a clear deviation, please list it as a limitation in the introduction and include it in the discussion.

Line 35: Please clarify: "however, no data are available on live fish interacting with thrust wakes". Most of the previous studies (such as ref. 5 and 6) with real fish systems provide some data on live fish interacting with thrust wake such as kinematic data representative for the hydrodynamic interactions.

Line 129: "with the foil in a stationary position", this needs to be clarified for a general reader. Should we interpret this as the foil being in a stationary position above the water? Otherwise, should we assume it is generating a von Karman drag wake with vortices? Since these airfoils have large-scale flow separation (laminar separation bubbles) at these sub-critical Reynolds numbers (below roughly Re = 100,000 / 200,000 depending on airfoil shape). Please clarify in the methods (and discussion in case there may be interference).

Figure 2: Should Panel C not be A, since it illustrates the phase lag calculation? Please resolve.

Figure 3: Shouldn't the legend for the color bar be Vorticity along the z-axis?

Line 202: "46% and 86% decrease compared to free-stream swimming" -- does this mean there are only two cases analyzed here? How did you establish these values? What is the zone around the fish head you used to evaluate pressure to make the comparison? Does this limit the analyses? Please clarify in a way the general reader can understand.

Line 323: You claim "swimming in-line is a frequently occurring situation" -- how often do fish swim in line compared to other formations?

[Editors' note: further revisions were suggested prior to acceptance, as described below.]

Thank you for resubmitting your article entitled "In-line swimming dynamics revealed by fish interacting with a robotic mechanism" for further consideration by *eLife*. Your revised article has been evaluated by a Reviewing Editor, who has drafted the below feedback to guide final revisions. A Senior Editor has overseen the re-evaluation.

The article has been improved but there are some remaining issues that need to be addressed:

The authors need to fully address the helpful feedback from the reviewers on the limitations of their hydrodynamic analysis. NACA 0012 is not an appropriate Low-Reynolds number airfoil and suffers from laminar separation bubbles at low Reynolds numbers, a low Reynolds number drag-crisis resulting in high drag, and a stall hysteresis loop. Classic low Reynolds number airfoil literature on these issues should not be ignored. That does not make the work unpublishable, but the reader needs to know about these limitations regardless of the fact that indeed many previous papers ignored them -- so that future studies can also consider more appropriate airfoils to see if that makes a difference to conclusions. Demonstrating that this has indeed an insignificant effect in the present research paradigm would require comparing a sub- and supercritical airfoil and showing that the outcome is highly similar; especially considering the lift-drag ratio of NACA 0012 is so much lower than even a flat or cambered plate at a low Reynolds number.

Also, while wall effects may be acceptable for answering the present research question, given how close the walls are, there needs to be a more detailed discussion of possible limitations. Wall interference is characterized based on airfoil/ propulsive body length, not width. For largely separated flow phenomena, such as shed vortices, there can be a marked wall effect when they are this close. It may not invalidate the work, but the cited literature ("In addition, wall effects (Webb, 1993) have minor to no measurable effects on fish swimming in our experiments.") lacks the fluid mechanic depth to argue that there is no measurable effect. That said, the experiment does seem representative of fish swimming close to a solid boundary, and for those conditions, the findings appear robust. There is no reason to believe that the non-quantified wall effect would disqualify the work for understanding how fish swim further away from a wall. But *eLife* offers the space to report that wall effects were not quantified (in the way a fluid mechanist would expect this to be done; boundary layer thickness at the wall is not actually sufficient for this). Given how close the walls are in body lengths this requires more consideration in future work, for example, with a robot fish mounted on a force sensor. The authors do not need to do this work for inclusion in the present article, if claims are tempered appropriately.

Addressing the above points should not be much work, but will improve the rigor of future fish swimming research. The Reviewing Editor has kindly offered to answer any questions you may have about these revision requests.

---

## [Author Response]

Essential revisions:1) In the summary of potential energy-saving mechanisms in schooling fish, please include the "vortex-phase matching" mechanism (ref. 33 in the references); it is related to the mechanism found in this paper, as discussed in the Discussion section (Line 291).

We thank the reviewers for this suggestion, and we have added a new analysis, figure, supplemental movie, and discussion of this important point. In our revision we now included an analysis of the phase difference between the flapping hydrofoil and the fish swimming in its wake (Line 186-198). We found a linear relationship between the fish-foil distance and the phase difference (see new Figure 3 / supplemental video 3). This supports previous findings using robotic systems that have studied vortex-phase matching, and demonstrates that live fishes can vortex phase-match. We believe that this is the first demonstration of this phenomenon.

As suggested, we accordingly mention the vortex-phase matching mechanism (Li et al., 2020) in the first paragraph (L327-330) of the section on “Swimming efficiency in thrust wakes” (discussion). We wish to note, however, that in the Li et al. paper, the vortex phase matching occurs in fish swimming nearly side-by side, and not in the in-line configuration, so our experiments here address another aspect of vortex phase matching where fish swim in a tandem arrangement.

2) The justification for the effectiveness of the rigid NACA airfoil is not clear. In Line 108, you mentioned "similar hydrodynamic characteristics", but what are these characteristics? In Line 125 and the Materials and methods, you mentioned similar reverse Karman vortex streets and increased speeds in the wake. A quantitative comparison would help justify and quantify the similarity. Further, we are missing a discussion of the Reynolds number regime. NACA airfoils are super-critical airfoils, meaning that they have been designed for high Reynolds numbers during which laminar flow separation is a minor issue. Operating such super-critical airfoils at sub-critical Reynolds numbers usually results in large-scale flow separation, such as laminar separation bubbles, poor stall behavior, and a drag crisis. Variable camber and a 3D shape (a fish-like body) or sharp leading edge of a 2D airfoil (optimal sub-critical airfoil shape) can dramatically reduce the issues. This is well established in the low Reynolds number aerodynamics literature and applies 1:1 to hydrodynamics, but is often overlooked by biomechanists working on swimming. Please outline the issues as limitations of the study in the methods and Discussion sections.

Thank you for pointing out where we were unclear on the choice of the NACA 0012 hydrofoil and the missing discussion of the Reynolds number regime. We would like to emphasize that the NACA 0012 shape specifically is commonly used to generate fish-like wakes in studies of biomimetic propulsion and to emulate a fish body shape (Akhtar et al., 2007; Karbasian and Esfahani, 2017; Lauder et al., 2007, 2011; Shua et al., 2007; Van Buren et al., 2019; Zhang et al., 2019). Furthermore we revised the manuscript text (L121-122, L141-142) to better explain what we mean by “similar hydrodynamic characteristics”. As shown in previous work, fish wakes are characterized by their reverse Karman vortex street and increased average flow speeds in the wake. In our work, we focused on the geometry of the wake, i.e., similar relative axial and lateral spacing between vortices while maintaining a similar width wake in lateral direction (≈4cm). These properties are well defined via the Strouhal number and the width of the wake. We now report both Strouhal numbers of fish and foil as well as the wake widths (L460-462). In addition, we believe that our visualization of the wake in Figure S2 supports our claim of similarity. For completeness we also report the Reynolds number for the flapping foil as Re = ULν=20100, U=0.3ms, L = 6.7cm (cord length), ν=10−6m2s (L463-464). Given these data, we are confident that the foil is operating as expected and the issues related to laminar separation bubbles, poor stall behavior and drag crisis can be excluded.

3) The novelty of the robotics-based experimental approach is overstated, given that several earlier studies have investigated the response of live fish to thrust wakes, generated by pitching airfoils or robotic fish. Please give due credit to previous studies and temper claims accordingly. Such strong novelty claims are not needed to make a compelling case for publication in eLife -- our policy is to focus on informative scientific contributions in the context of the literature.

We appreciate the reviewer’s comment and have toned down the novelty of the robotics-based approach by including the following papers (L118-120):

Fish in the thrust wake of a hydrofoil (Harvey et al., 2022; Zhang et al., 2019)Fish in the thrust wake of robotic fish model (Marras and Porfiri, 2012)

However, we do wish to add that there are very few papers on live fish swimming in a thrust wake, and almost all papers focus on fish in a drag wake in which energy savings would be expected. Here we show that energy savings can occur when fish swim in a thrust wake using pressure computations on the surface of freely-swimming fish, an approach that has not been previously undertaken.

4) It seems like the length of the test section in the experiments is on par with the body length of the animals, hence wall-effects could confound the study to some degree. Please use the established literature and any estimation within your reach to outline wall-effects in the methods, and discuss the results in light of this limitation. Provide guidance for follow-up studies on how they could improve on the current experimental design.

We have added text to the methods section in the manuscript to explain our view on wall effects in these experiments (which we believe to be minimal) (L437-441). The recirculating flow tank and experimental apparatus used for the experiments described in this manuscript have been used for more than 50 previous papers on fish locomotion some of which also use the flapping foil propulsion system to generate wakes. Below we provide some key citations and reviews to this previous work, and here we respond directly to the comment on fish size relative to flow tank dimensions. The test section is 30 cm square in cross section and over a meter long but is open at both the upstream and downstream ends to allow flow to enter and leave so blocking effects in this dimension are not an issue. The overall length of the tank in the longitudinal dimension is two meters. Trout in our experiments varied in both length from 15 to 20 cm, and were 3 cm or less in body width. Thus, the side-to-side dimensional comparison of tank width to trout width has a ratio of 10:1. The relevant literature on this topic (see below) indicates that corrections for fish blocking of flow are not needed for less than 10%, although these studies have not quantified flow patterns. But we have previously done this by using PIV to assess flow across the entire width of the flow tank, and did not observe any measurable effect of fish on flow in the far field even with the tail beating from side to side as vortices are rapidly convected downstream.

Also, since we have previously quantified the boundary layer thickness in this same tank (Tytell and Lauder, 2004) which was approximately 5 mm at the flow speeds of these experiments, and fish swam in the center of the tank, we do not believe that wall effects have any measurable effect on our results.

Some key references:

Tytell, E. D. and Lauder, G. V. (2004). The hydrodynamics of eel swimming. I. Wake structure. Journal of Experimental Biology 207, 1825-1841.

Webb, P. W. (1993). The effect of solid and porous channel walls on steady swimming of steelhead trout Oncorhynchus mykiss. Journal of Experimental Biology 178, 97-108.

Kline, R. J., Parkyn, D. C. and Murie, D. J. (2015). Empirical modeling of solid-blocking effect in a Blazka respirometer for gag, a large demersal reef fish. Adv. Zool. Bot 3, 193-202.

Lauder, G. V. (2006). Locomotion. In The Physiology of Fishes, Third Edition, (D. H. Evans and J. B. Claiborne, eds.), pp. 3-46. Boca Raton: CRC Press.

Lauder, G. V. (2015). Fish locomotion: recent advances and new directions. Annual review of Marine Science 7, 521-545.

Lauder, G. V. and Tytell, E. D. (2006). Hydrodynamics of undulatory propulsion. In Fish Biomechanics. Volume 23 in Fish Physiology, (R. E. Shadwick and G. V. Lauder, eds.), pp. 425-468. San Diego: Academic Press.

5) The role of 3D effects needs to be elaborated; the 2D experimental data limit the generality and conclusions on actual fish collective behavior. Please include these issues in the introduction, explain how you converged on this experimental design in the methods, and discuss the limitations in the discussion and how future studies could expand the current work to the full 3D realm. In the discussion, please cite papers comparing 2D and 3D wakes to infer any possible implications for interpreting the quantitative measurements of vorticity and pressure in the paper, and extrapolating these findings to 3D fish swimming close together. The merit of 2D studies is not questioned, but does limit the conclusions and should inspire future 3D studies to further test your conclusions and extrapolate the implications for actual (3D) fish collective behavior.

Thank you for your comment. We followed your suggestion and explicitly mention in the first paragraph of the results (L142-145) that we are employing a 2D experimental approach, and we discuss the limitations of our 2D experimental approach with respect to 3D effects in a dedicated paragraph in the discussion (L382-398).

In addition, we would like to emphasize in experiments such as these, in which fish swim in-line with a robotic flapping foil, that limiting the height of the foil (to effectively make the experiments more 3D) is quite challenging as fish are much less likely to voluntarily hold position due to the smaller foil dimensions in our relatively large experimental space. We do agree that follow-up experiments with a shorter, more 2D foil, would be valuable indeed, but believe that the data we present with a longer foil that makes live fish data acquisition more feasible, is a good first step.

6) Please briefly comment on other sensory modalities (touch, the vestibular system, vision), citing the relevant literature, before going in-depth on which senses play a role, so general readers (across biology and engineering) can understand your framework and its justification/limitations. Expanding the introduction and discussion on sensory modalities beyond the chosen/motivated focus will help readers understand your scientific reasoning.

Thank you for this suggestion. We added new text mentioning sensory modalities in the introduction (L112-115).

Fish moving in fluids use (1) vision, (2) the lateral line, and (3) the vestibular system to control their body motion. All of these modalities have been the subject of numerous studies over the years. It’s a lot to add a discussion of the various modalities and what they are used for to this manuscript, but we did add (L115-116) information to the effect that fish in our experiments had all modalities available to them for flow detection, a very basic description of the three modalities involved, and added some key references to each of the modalities as indicated below here:

Coombs, S. and Montgomery, J. (2014). The Role of Flow and the Lateral Line in the Multisensory Guidance of Orienting Behaviors. In Flow Sensing in Air and Water, pp. 65-101: Springer.

Platt, C. (1973). Central Control of Postural Orientation in Flatfish II. Optic-Vestibular Efferent Modification of Gravistatic Input. Journal of Experimental Biology 59, 523-541.

Ali, M. (2013). Vision in fishes: New approaches in research: Springer Science and Business Media.

7) In the methods section, please provide detailed reasoning for the selection of particle image velocimetry plane chosen for the computation of vorticity from 2D data and inference of pressure from 2D data. Although the cited paper from Dabiri is helpful, a summary of the approach for pressure inference is essential to clarify its reliability. This knowledge cannot be assumed to be generally known among the readership.

Thank you for this comment. We added a summary of the approach to compute pressure fields from 2D velocity fields (L536-542). In brief, we used the standard fish-PIV horizontal plane (common to many investigations over recent decades) as a basis for estimating pressures and forces, and we have used and validated this approach (against measured forces generated by a flapping foil) in previous publications (cited below, L544-545), and added these citations at this location to assist readers.

Thandiackal, R. and Lauder, G. V. (2020). How zebrafish turn: analysis of pressure force dynamics and mechanical work. The Journal of Experimental Biology 223, jeb223230.

Thandiackal, R., White, C. H., Bart-Smith, H. and Lauder, G. V. (2021). Tuna robotics: hydrodynamics of rapid linear accelerations. Proceedings of the Royal Society B: Biological Sciences 288, 20202726.

Lucas, K. N., Dabiri, J. O. and Lauder, G. V. (2017). A pressure-based force and torque prediction technique for the study of fish-like swimming. PLoS ONE 12, e0189225.

Lucas, K. N., Lauder, G. V. and Tytell, E. D. (2020). Airfoil-like mechanics generate thrust on the anterior body of swimming fishes. Proceedings of the National Academy of Sciences, 201919055.

8) Please clarify if the flow is laminar or turbulent in the tested flow regime and add a plot showing at what distance the switch between double vortex split and single-sided vortex occurs.

We now report the Reynolds number (Re=20100) in the ‘Flapping foil’ section in the Materials and methods (L463-464) and clarify that we are operating in a turbulent flow regime (Please also see our response to comment 2). And please see our response to comment 11 regarding the critical distance for double-sided vs single-sided vortex interactions.

9) Please carefully double-check the accuracy of wording in "increase in flow speed in the thrust wake" in the abstract and introduction, which may be in contrast with flow refuging behind bluff bodies.

We confirm the accuracy of the wording. Flow refuging is related to drag wakes, e.g., behind bluff bodies whereas we observe increased average flow speeds in thrust wakes. In this case, we have *accelerated* the flow beyond the free-stream using our robotic flapper that generates a fish-like reverse Karman vortex street.

10) In Figure S2, it appears that the fish and airfoil differ, yet they appear to shed similar vortex pairs -- is this the case? If so, please discuss this matter in your experimental design (and discussion where relevant).

We apologize for the confusion here, but the airfoil and the fish do indeed generate similar wakes! This has actually been used in previous studies that use flapping foils as models for fish-like propulsion. The flapping foil serves as a model for the tail fin portion of the fish body. Given that the fish body and its tail are flexible they have different kinematics than the rigid flapping foil. Nonetheless, by purposefully defining the kinematics of the rigid flapping foil we can achieve similar excursions of the tail tip and ultimately generate similar vortex pairs. We added more explanations in the ‘Flapping foil’ section in the methods (L446-451) as well as in the caption of Figure S2. Please also see our response to comment 13. Also, the use of a flapping foil model system to generate fish-like wakes is quite established in the fish bio-fluids literature (both computational and experimental) and we cite several relevant papers in the revised manuscript.

11) Line 158: The "one-sided" versus "two-sided" vortex interaction analyses are qualitative. Is it possible to quantitatively analyze the relationship between the phase of the airfoil and the phase of head movements, since the thrust wake is linked to the phase of the airfoil? In Line 184, you mentioned fish adopt two-sided or one-sided interactions depending on the lateral offset and the distance between the airfoil and real fish. Is it possible to estimate the critical distances which drive the fish to switch from two-side vortex interaction to one-side?

Many thanks to the reviewers for their suggestions to further investigate phase relationships, as we believe that this has significantly improved the manuscript. As mentioned in our response to comment 1, we have added a quantitative analysis (L186-198) of the phase difference between the hydrofoil and fish swimming in its wake (see the new figure and the new supplemental movie). As a result, we found that the phase difference varies linearly with the distance between fish and foil. We agree that the analysis of double-sided vs single-sided vortex interactions can be considered mostly qualitative at this point, although it is based on the quantification of the flow field via PIV. To estimate a critical distance for the transition between these interaction modes, we believe that more data would be needed. Unfortunately, it is not trivial to obtain a large enough sample size of PIV data capturing live fish swimming in the laser sheet while performing our investigated behavior, and thus we think that estimating the critical distance is out of the scope of this work. From the data we have, we predict that double-sided vortex interactions require a very good in-line alignment with the foil to be able to capture vortices at both sides of the body, and we predict a critical distance within one body width from the foil center line.

12) Since fish might also swim further away (e.g., S Movie 2), what are the parameters you used to define/filter in-line swimming patterns? Line 165: You state "time their movements accordingly" to interact with the wakes generated by the airfoil -- but is it possible to add quantitative analyses of how the fish may time their movements? For example, by determining the phase difference between the airfoil motion and fish head oscillation? This seems also applicable to Line 142. Finally, in Line 146, please explain to the general reader why you did not consider using fish body wavelength for determining phase lag, which is more typical for fish kinematics. If this is not possible, please justify your processing choices accordingly, so the general reader can understand your choices and any limitations this may have for interpreting the findings.

For our analysis we considered as many swimming trials as possible that showed prolonged in-line swimming (lasting 5 to 10 seconds and more). As a result, we captured this behavior over a range of distances between fish and foil as trout naturally chose to swim at different distances. This allowed us, as suggested by the reviewers, to now include an analysis of the phase difference for different distances (see also response to comment 1). We found a linear relationship, and the parameters of the linear fit support the hypothesis of vortex interaction and synchronization (L186-198).

Thank you for mentioning the body wavelength metric. The overall phase lag along the body and the body wavelength carry similar information and can be computed one from the other (λ=2πΔΦ). Whereas phase lags are commonly reported in the field of neuroscience and robotics, wavelengths appear more often in biomechanics. To reach a larger audience we now report the wavelength in addition to the phase lag in the methods and the supplementary data file (L507-509).

13) Line 122 states you chose St around 0.267. Please outline how you selected the oscillation frequency of the airfoil with respect to the natural tail beat frequency. Also, please clarify how you selected the experiment parameters a_{sway}, a_{yaw}, amplitude, since there are different combinations of these parameters possible that give St ~ 0.267, and not all combinations are biologically representative or informative. Please include typical kinematics data of real fish here to help justify the parameter selection and, whenever there is a clear deviation, please list it as a limitation in the introduction and include it in the discussion.

We would like to refer the reviewers to our section ‘Flapping foil’ in the Materials and methods. Here we reported all the parameters including a_{sway} and a_{yaw}. We added additional explanations (L461) regarding the choice of these amplitudes (width of the wake) and the frequency (in the range of natural tail beat frequencies). Regarding the typical kinematics of real fish, we would like to point out that our goal was not to match the kinematics of the tail fin of real fish with our flapping foil. Our aim was rather to induce similar wakes with the rigid flapping foil by achieving fish-like Strouhal numbers together with the corresponding width of the wake (≈4cm). The results are illustrated in Figure S2, where we provide a direct comparison of the resulting thrust wakes.

Also, in quite a number of previous publications over the years we have published both kinematics and hydrodynamic wakes from swimming fishes and so we prefer not to duplicate that information here. The thrust wake generated by the flapping NACA 0012 very closely matches fish wakes produced by the tail. In addition, just last year we published swimming kinematics from 44 species of fishes (DiSanto et al., 2021) and provide a large data table with waveform characteristics and Strouhal numbers in addition to a number of other kinematic parameters including head and tail amplitudes for these 44 species of swimming fishes.

Di Santo, V., et al. (2021). "Convergence of undulatory swimming kinematics across a diversity of fishes." Proceedings of the National Academy of Sciences 118(49): e2113206118.

Line 35: Please clarify: "however, no data are available on live fish interacting with thrust wakes". Most of the previous studies (such as ref. 5 and 6) with real fish systems provide some data on live fish interacting with thrust wake such as kinematic data representative for the hydrodynamic interactions.

Thank you for your comment! We rephrased this sentence and modified the abstract accordingly (L36). We also added more references related to fish interacting with thrust wakes (Please see response to comment 3).

Line 129: "with the foil in a stationary position", this needs to be clarified for a general reader. Should we interpret this as the foil being in a stationary position above the water? Otherwise, should we assume it is generating a von Karman drag wake with vortices? Since these airfoils have large-scale flow separation (laminar separation bubbles) at these sub-critical Reynolds numbers (below roughly Re = 100,000 / 200,000 depending on airfoil shape). Please clarify in the methods (and discussion in case there may be interference).

We apologize for the possible confusion this term has caused. By stationary position we describe the state in which the hydrofoil is placed in the water but not moved. We have clarified this in the text now (L157). It is true that the stationary foil in the water generates a very small and narrow (due to the low drag of the NACA 0012 foil) Karman drag wake with small vortices. However, this happens on a scale several magnitudes smaller than the width of the fish tested in our experiments. In addition, we can assure the reviewer that we did not find laminar separation bubbles as the flow over the stationary foil is smooth at these Re. The main reason for having a reference condition with the foil in the water is to account for the behavioral aspect of fish interacting with any object whether it is moving or not. We wanted to have a good control condition that takes into account an object that fish could orient to regardless of its hydrodynamic properties. This allowed us to attribute the differences in fish swimming to hydrodynamic effects rather than any behavioral attraction between the fish and the foil.

Figure 2: Should Panel C not be A, since it illustrates the phase lag calculation? Please resolve.

It is correct that panel C illustrates the phase lags. However, we chose to order the panels as they appear in the text. In particular, we would like to keep panel A for the frequencies as this is the first result that we present and the order of panels is the same as in the text discussion.

Figure 3: Shouldn't the legend for the color bar be Vorticity along the z-axis?

Unfortunately, we are not entirely sure what the reviewer is referring to here. We do have a color bar on the right denoted with vorticity and the corresponding unit. The orientations are illustrated as well and are consistent with a right handed coordinate system (positive vorticity vector points out of the plane).

Line 202: "46% and 86% decrease compared to free-stream swimming" -- does this mean there are only two cases analyzed here? How did you establish these values? What is the zone around the fish head you used to evaluate pressure to make the comparison? Does this limit the analyses? Please clarify in a way the general reader can understand.

We have expanded the explanation of the computation of the head pressures in the Materials and methods section (L546-551). We now also include how we defined the zone around the fish that is used to derive the corresponding average values. It is correct that our analysis of the head pressures is at this point limited to two examples with double-sided vortex interactions and one example with single-sided vortex interactions. This is in part related to the difficulty of obtaining such samples, as fish have to swim in the laser sheet (for PIV) for multiple swimming cycles. Nonetheless, we believe that these data provide relevant insight into the underlying dynamics of fish swimming in the thrust wakes that are interesting for future modeling and experimental studies. We mention the limitations now explicitly in the discussion (L399-408).

Line 323: You claim "swimming in-line is a frequently occurring situation" -- how often do fish swim in line compared to other formations?

We appreciate the comment and have removed the word ‘frequently’ from the text in this paragraph (L411, L413) as this statement stems from our preliminary data on fish schools and needs further quantification. We have several ongoing fish schooling kinematic projects that will quantify exactly how often fish in schools position themselves in particular configurations, but at the moment we only have video data and no quantitative analyses.

References

Akhtar, I., Mittal, R., Lauder, G. V., and Drucker, E. (2007). Hydrodynamics of a biologically inspired tandem flapping foil configuration. In *Theoretical and Computational Fluid Dynamics* (Vol. 21, Issue 3, pp. 155–170). https://doi.org/10.1007/s00162-007-0045-2

Harvey, S. T., Muhawenimana, V., Müller, S., Wilson, C. A. M. E., and Denissenko, P. (2022). An inertial mechanism behind dynamic station holding by fish swinging in a vortex street. *Scientific Reports*, *12*(1), 12660.

Karbasian, H. R., and Esfahani, J. A. (2017). Enhancement of propulsive performance of flapping foil by fish-like motion pattern. In *Computers and Fluids* (Vol. 156, pp. 305–316). https://doi.org/10.1016/j.compfluid.2017.07.016

Lauder, G. V., Anderson, E. J., Tangorra, J., and Madden, P. G. A. (2007). Fish biorobotics: kinematics and hydrodynamics of self-propulsion. *The Journal of Experimental Biology*, *210*(Pt 16), 2767–2780.

Lauder, G. V., Lim, J., Shelton, R., Witt, C., Anderson, E., and Tangorra, J. L. (2011). Robotic Models for Studying Undulatory Locomotion in Fishes. In *Marine Technology Society Journal* (Vol. 45, Issue 4, pp. 41–55). https://doi.org/10.4031/mtsj.45.4.8

Li, L., Nagy, M., Graving, J. M., Bak-Coleman, J., Xie, G., and Couzin, I. D. (2020). Vortex phase matching as a strategy for schooling in robots and in fish. *Nature Communications*, *11*(1), 5408.

Marras, S., and Porfiri, M. (2012). Fish and robots swimming together: attraction towards the robot demands biomimetic locomotion. *Journal of the Royal Society, Interface / the Royal Society*, *9*(73), 1856–1868.

Shua, C., Liua, N., Chewa, Y., and Lub, Z. (2007). Numerical simulation of fish motion by using lattice Boltzmann-Immersed Boundary Velocity Correction Method. In *Journal of Mechanical Science and Technology* (Vol. 21, Issue 9, pp. 1352–1358). https://doi.org/10.1007/bf03177420

Van Buren, T., Floryan, D., Bode-Oke, A. T., Han, P., Dong, H., and Smits, A. (2019). Foil shapes for efficient fish-like propulsion. In *AIAA Scitech 2019 Forum*. https://doi.org/10.2514/6.2019-1379

Zhang, P., Krasner, E., Peterson, S. D., and Porfiri, M. (2019). An information-theoretic study of fish swimming in the wake of a pitching airfoil. In *Physica D: Nonlinear Phenomena* (Vol. 396, pp. 35–46). https://doi.org/10.1016/j.physd.2019.02.014

[Editors' note: further revisions were suggested prior to acceptance, as described below.]

The article has been improved but there are some remaining issues that need to be addressed:The authors need to fully address the helpful feedback from the reviewers on the limitations of their hydrodynamic analysis. NACA 0012 is not an appropriate Low-Reynolds number airfoil and suffers from laminar separation bubbles at low Reynolds numbers, a low Reynolds number drag-crisis resulting in high drag, and a stall hysteresis loop. Classic low Reynolds number airfoil literature on these issues should not be ignored. That does not make the work unpublishable, but the reader needs to know about these limitations regardless of the fact that indeed many previous papers ignored them -- so that future studies can also consider more appropriate airfoils to see if that makes a difference to conclusions. Demonstrating that this has indeed an insignificant effect in the present research paradigm would require comparing a sub- and supercritical airfoil and showing that the outcome is highly similar; especially considering the lift-drag ratio of NACA 0012 is so much lower than even a flat or cambered plate at a low Reynolds number.

Previous work (Triantafyllou et al. 1993; Triantafyllou et al. 2004; Anderson et al. 1998) suggests that NACA 0012 foils are appropriate for the purpose of our study where we aim to create fish-like wakes. They demonstrate that these foils show high propulsive efficiencies if they are operated in a Strouhal number range of 0.25 to 0.35 and that the reverse Karman vortex street with its increased wake velocities is related to thrust production. In part, high efficiency is associated with leading-edge vortices that are convected downstream and interact with trailing-edge vortices that result in a reverse Karman street (Anderson et al. 1998). Furthermore, previous studies have shown that fish operate in a similar range of Strouhal numbers and also produce reverse Karman vortex streets (Saadat et al. 2017).

We acknowledge the points raised by the reviewer regarding NACA 0012 being a supercritical airfoil and that a comparison to subcritical foils (cambered, flat) could be interesting. We mention this in the Materials and methods section now. However, as mentioned above, we were able to generate thrust producing wakes characterized by reverse Karman vortex streets with increased wake velocities which were sufficient for the purpose of our experiments as they provide the flow structures that fish face in wakes of other fish.

The NACA 0012 foil, moved in the manner we programmed, generates an excellent fish-like wake. We are well aware of separation bubbles formed by NACA 0012 airfoils (see our previous paper using static testing of these airfoils which shows such bubbles (Domel et al., 2018)). Many of these studies (like this one of ours) are performed under static conditions with the airfoil stationary at each angle of attack tested. Under these conditions and at relatively low Re of 5000 to 100,000 a separation bubble will indeed form and then can peel off causing a loss of lift if it grows to a large size.

But our goals in this manuscript were to (1) use a fish-shape to (2) generate a fish-like wake. The NACA 0012 foils are excellent for this purpose as we show in this and a number of previous papers quantifying the wakes of these foils moved in heave, pitch, and both. We emphasize that in our experiments the airfoil is *moved* and that it does not matter that under these dynamic conditions that a separation bubble forms. Any separation bubble becomes incorporated into the overall wake vortex structure that is shed as the foils move back and forth.

We cite one of our previous papers with NACA 0012 foils below (Lauder et al., 2007) to show PIV data from heaving and pitching NACA 0012 foils. Movement of the airfoil in the manner that we programmed generates a fish-like wake that was the whole point of using the flapping foil.

Anderson, J. M., Streitlien, K., Barrett, D. S., and Triantafyllou, M. S. (1998). Oscillating foils of high propulsive efficiency. Journal of Fluid Mechanics, 360, 41–72.

Triantafyllou, G. S., Triantafyllou, M. S., and Grosenbaugh, M. A. (1993). Optimal Thrust Development in Oscillating Foils with Application to Fish Propulsion. Journal of Fluids and Structures, 7(2), 205–224.

Triantafyllou, M. S., Techet, A. H., and Hover, F. S. (2004). Review of experimental work in biomimetic foils. IEEE Journal of Oceanic Engineering, 29(3), 585–594.

Saadat, M., Fish, F. E., Domel, A. G., Di Santo, V., Lauder, G. V., and Haj-Hariri, H. (2017). On the rules for aquatic locomotion. In Physical Review Fluids (Vol. 2, Issue 8). https://doi.org/10.1103/physrevfluids.2.083102

Domel, A. G., Saadat, M., Weaver, J., Haj-Hariri, H., Bertoldi, K. and Lauder, G. V. (2018). Shark denticle-inspired designs for improved aerodynamics. Journal of the Royal Society Interface 15, 20170828.

Lauder, G. V., Anderson, E. J., Tangorra, J. and Madden, P. G. A. (2007). Fish biorobotics: kinematics and hydrodynamics of self-propulsion. Journal of Experimental Biology 210, 2767-2780.

Also, while wall effects may be acceptable for answering the present research question, given how close the walls are, there needs to be a more detailed discussion of possible limitations. Wall interference is characterized based on airfoil/ propulsive body length, not width. For largely separated flow phenomena, such as shed vortices, there can be a marked wall effect when they are this close. It may not invalidate the work, but the cited literature ("In addition, wall effects (Webb, 1993) have minor to no measurable effects on fish swimming in our experiments.") lacks the fluid mechanic depth to argue that there is no measurable effect. That said, the experiment does seem representative of fish swimming close to a solid boundary, and for those conditions, the findings appear robust. There is no reason to believe that the non-quantified wall effect would disqualify the work for understanding how fish swim further away from a wall. But eLife offers the space to report that wall effects were not quantified (in the way a fluid mechanist would expect this to be done; boundary layer thickness at the wall is not actually sufficient for this). Given how close the walls are in body lengths this requires more consideration in future work, for example, with a robot fish mounted on a force sensor. The authors do not need to do this work for inclusion in the present article, if claims are tempered appropriately.

Wall effects is actually a topic that we have previously investigated in some detail using the approach recommended above: using an experimental system with force measurement to quantify locomotor forces during propulsion at varying distances from the walls of the flow tank, and with both kinematic and PIV analyses of flow patterns near the wall. One relevant paper is Quinn et al. (2014a, citation below) and this study was actually performed in the *same* flow tank and with the *same* robotic flapper used for this manuscript. We used a 6-axis force/torque sensor to quantify propulsion speed, forces, and efficiency at varying distances from the wall.

The results of this analysis, discussed below in some detail, show why we are very confident that we are not experiencing significant wall effects in our current manuscript.

In brief, here are the results from that previous study Quinn et al. (2014a) paper to illustrate our key response points. Quinn et al. studied the propulsion of relatively large flexible panels (150mm span by 195 mm length) at varying distances from both the wall and bottom of the flow tank. Please note that this panel size is much larger in area than the trout studied here and that the panels, when moved, came as close as 15mm from the wall which is much, much closer than the trout studied here which were in the middle of the tank (approx. 30 cm on a side and 1 meter long working area for filming). Plastic panels of three flexural stiffnesses were studied (see Table 1 Quinn et al. (2014a)) and these panels were moved at the leading edge (in a manner similar to the foil motion used in this manuscript) to generate a propulsive wave. The flexural stiffness of the panels in Quinn et al. (2014) was specifically chosen to encompass the range of actual fish flexural stiffness which range from 10^–3^ to 10^-6^ (Shelton et al., 2014) and Panel B (Table 1 Quinn et al. (2014a) ) matches nicely the flexural stiffness of trout bodies.

Now, Quinn et al., 2014a: Figure 5 shows the effect on propulsive speed and economy at different distances from the wall, given as the parameter d/a where d=mean distance from the wall, and a=heave amplitude of the flexible body at the leading edge. Note that there is minimal to no effect of distance on economy and little to no effect of distance on self-propelled speed for all panels at d/a >5. Please note also that for the most trout-like panel (B) there is almost no effect at any distance.

In this manuscript, trout operated at a value of between 7 and 3.5 depending on whether “a” is taken as head oscillation or tail oscillation amplitude.

Although swimming close to a surface with an undulatory body can certainly improve propulsive efficiency and alter the time-dependent profiles of forces and torques, we believe that these previous experiments using the *exact same* experimental system show that it is highly unlikely that wall effects have influenced our results for trout swimming in the center of the flow tank.

One final study from Quinn et al. (2014b) used another flapper/flow tank system at Princeton Univ. to investigate the wall effect for a rigid pitching foil at different distances from the wall. Their data show that for conditions similar to that of our trout experiments where trout are 15 cm from the wall, that there is no effect on propulsive forces.

Overall, we would like to emphasize as these studies show, that the motion of flexible bodies near surfaces (particularly under conditions of self-propulsion where free-stream flow convects vortices downstream away from the undulatory surface) has some unexpected results which are not well represented by classical static airfoils studied near a wall.

One last point. A second study from our lab (Blevins and Lauder, 2013) also analyzed ground/wall effects using a flexible propulsor (again in the *same* flow tank and with the *same* robotic flapper system). That study quantified self-propelled speed, flow patterns near the wall with PIV, and kinematics, and calculated the cost of transport (using data from a force transducer) comparing undulatory propulsion in the center of the flow tank to a position where the undulatory body approached within 1 cm of the wall. This study found some small differences in the cost of transport at some actuation frequencies, but that at least two actuation patterns show no difference.

Again, the trout in our experiments swam in the center of the flow tank, and not near any wall. So from these results also we expect that the propulsion of our trout was not in any way altered by a “wall effect”.

Quinn, D. B., Lauder, G. V. and Smits, A. J. (2014a). Flexible propulsors in ground effect. Bioinspiration and Biomimetics 9, 1-9.

Quinn, D. B., Moored, K. W., Dewey, P. A. and Smits, A. J. (2014b). Unsteady propulsion near a solid boundary. Journal of Fluid Mechanics 742, 152-170.

Blevins, E. L. and Lauder, G. V. (2013). Swimming near the substrate: a simple robotic model of stingray locomotion. Bioinspiration and Biomimetics 8, 016005.

Shelton, R. M., Thornycroft, P. J. M. and Lauder, G. V. (2014). Undulatory locomotion by flexible foils as biomimetic models for understanding fish propulsion. Journal of Experimental Biology 217, 2110-2120.